# Bayesian Optimisation via Difference-of-Convex Thompson Sampling

## Abstract

Thompson sampling is a method for Bayesian optimisation whereby a randomly drawn belief of the objective function is sampled at each round and then optimised, informing the next observation point. The belief is typically maintained using a sufficiently expressive Gaussian process (GP) surrogate of the true objective function. The sample drawn is non-convex in general and non-trivial to optimise. Motivated by the desire to make this optimisation subproblem more tractable, we propose *difference-of-convex Thompson sampling* (DCTS): a scalable method for drawing GP samples that combines random neural network features with pathwise updates on the limiting kernel. The resulting samples belong to the *difference-of-convex* function class, which are inherently easier to optimise while retaining rich expressive power. We establish sublinear cumulative regret bounds using a simplified proof technique and demonstrate the advantages of our framework on various problems, including synthetic test functions, hyperparameter tuning, and computationally expensive physics simulations.

## 1 Introduction

Bayesian optimisation (BO) considers the problem

$$\max_{\boldsymbol{x} \in \mathcal{X}} f(\boldsymbol{x}), \tag{1}$$

where $\boldsymbol{x} \in \mathcal{X}$ is a $d$-dimensional input vector in a compact domain $\mathcal{X} \subset \mathbb{R}^d$, with $f : \mathcal{X} \to \mathbb{R}$ a real-valued, generally nonconvex objective function. Typically in BO, $f$ is treated as a black-box function with no available gradient information and is assumed to be expensive to evaluate. Examples in science and engineering often treat $f$ as the output of a computationally expensive physics simulation (Shahriari et al., 2016), while in machine learning, $f$ frequently represents tasks like algorithm optimisation, such as hyperparameter tuning (Snoek et al., 2012; Klein et al., 2017; Chen et al., 2022). Consequently, our goal is to find a good solution to (1) using a limited number of $f$ evaluations, necessitating an optimisation policy that balances exploration and exploitation.

To that end, BO iteratively constructs a probabilistic surrogate model, $\mathsf{f}_t$, typically a Gaussian process (GP), to approximate the true objective function $f$. Assume that observations of $f$ are given by $y_t = f(\boldsymbol{x}_t) + \epsilon_t$, where each $\epsilon_t \sim \mathcal{N}(0, \epsilon^2)$ is independent and identically distributed (i.i.d.). As data pairs $(\boldsymbol{x}_t, y_t)$ are collected, the updated GP posterior represents the belief about $f$ on $\mathcal{X}$. At each iteration $t$ of BO, an *acquisition function* $\alpha_t(\boldsymbol{x})$ is defined to measure the utility of obtaining new observation data for any input. More precisely, the subproblem

$$\boldsymbol{x}_t \triangleq \arg\max_{\boldsymbol{x} \in \mathcal{X}} \ \alpha_t(\boldsymbol{x}) \tag{2}$$

is defined at each round, then the objective function is evaluated at $\boldsymbol{x}_t$ to get an observation $y_t$, and this new observation is used to update the GP posterior.

Thompson sampling (TS) is a popular stochastic algorithm for BO that balances exploration and exploitation by simply taking $\alpha_t(\boldsymbol{x})$ to be a sample of the GP surrogate posterior. While most theoretical regret guarantees

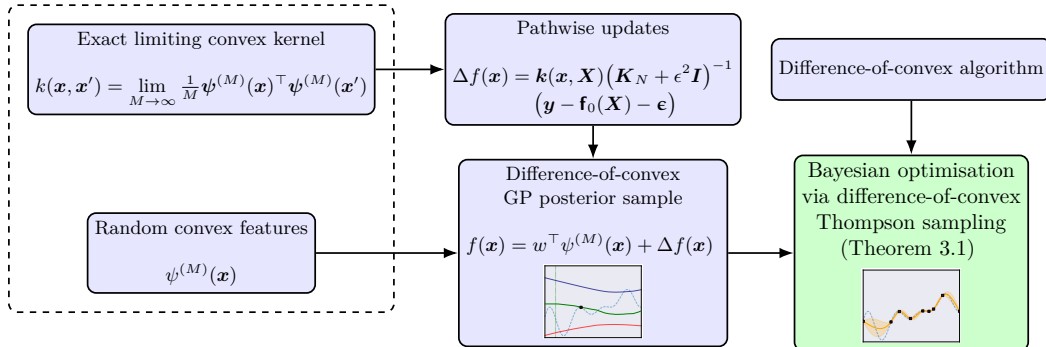

Figure 1: A GP posterior sample is generated using random features and their exact limiting kernel (derived analytically), which is then optimised via the difference-of-convex algorithm. Sample = convex component + concave component.

for Thompson sampling and other acquisition functions require (2) to be solved globally and exactly, in practice this is difficult to guarantee due to the typically non-convex nature of $\alpha_t(\boldsymbol{x})$ (Balandat et al., 2020, § F.1). In practice, most acquisition solving methods can only guarantee local solutions. Strategies for solving (2) include DIRECT (Jones et al., 1993), global combinatorial search algorithms (which do not scale well), or multi-start methods combined with local optimizers (e.g., L-BFGS) (Wilson et al., 2018; Do et al., 2024; Balandat et al., 2020). Regardless, solving (2) remains a significant computational challenge that is frequently overlooked in the design of BO algorithms.

## 1.1 Contributions

Successful BO algorithms generally require two things: judicious choice of kernel and kernel hyperparameters Garnett (2023, § 10.7), and accurate acquisition function optimisations. Here, we address the latter by constructing the GP prior such that the GP posterior samples belong to a class of functions called *difference-of-convex* (DC), making them more amenable to local optimisation. We outliune this process in Figure 1.

- We introduce *difference-of-convex Thompson sampling* (DCTS), a framework for constructing GP posterior samples that naturally admit a DC decomposition. This enables more efficient and effective local optimisation of (2) in the context of Thompson sampling.

- Under the setting of random features and pathwise updates, we establish sublinear cumulative regret bounds for Thompson sampling. Our use of a path-wise update for posterior sampling leads to more streamlined proofs compared to those found in similar literature and may be of independent interest.

- We show experimentally that exploiting this DC structure enables more effective optimisation of the acquisition function, leading to overall better performance compared to alternative approaches. This is validated across a variety of problems, including neural network hyperparameter tuning, benchmark functions, and computationally expensive physics simulations.

Our method specifically targets the way in which Thompson sample paths are drawn and optimised, and can be incorporated into any broader BO algorithms which utilise Thompson sampling.

## 1.2 Related work

The tractability of (2) has been approached from multiple angles, including submodularity (Wilson et al., 2018) and Lipschitz continuity (Mutny & Krause, 2018). Balandat et al. (2020, §4) describe how this problem is handled practically with the BoTorch package. By default, the package uses L-BFGS-B in conjunction with an initialisation heuristic that exploits fast batch evaluation of acquisition functions. The authors acknowledge that existing literature on the convergence of acquisition function optimisers is limited.

Wang et al. (2023) provides a recent survey on Bayesian optimisation, which includes advances in high-dimensional BO, multi-fidelity approaches, and parallel/batch BO. However, the authors highlight the intractability of (2).

Ament et al. (2023) introduce a new acquisition function for expected improvement which improves the solvability of (2) by addressing the vanishing gradient problem. Xie et al. (2024) use piecewise linear approximations on the kernel to cast (2) as a mixed-integer program (MIP), which provides guarantees on its global solution. While global solutions of the acquisition are required to achieve sublinear regret bounds in BO, in practice they are rarely scalable with dimension, and most solving methods for (2) in the literature settle for a "good enough" local solution.

In a similar vein, our work aims to address this by inducing a favourable structure on $\alpha$ and subsequently exploiting it for higher quality local solutions.

## 2   Background

We provide a general overview of GPs for Bayesian optimisation (Section 2.1) and a summary of generic Thompson sampling. For a more detailed background, see Garnett (2023). Our notation is detailed in Section D.

### 2.1   Gaussian processes

Let $\mathcal{D} = (\boldsymbol{x}_i, y_i)_{i=1}^N$ be a training dataset of $N$ input-output pairs, where $y_i = f(\boldsymbol{x}_i) + \epsilon_i$. Here, $f$ is an unknown function, and $\epsilon_i \sim \mathcal{N}(0, \epsilon^2)$ represents i.i.d. measurement noise. A GP model assumes that any finite subset of function evaluations follows a joint Gaussian distribution. It is characterised by a mean function $\mu : \mathcal{X} \to \mathbb{R}$, which associates to every point $\boldsymbol{x}$ a mean parameter $\mu(\boldsymbol{x}) = \mathbb{E}[f(\boldsymbol{x})]$, and a positive semi-definite kernel function $k : \mathcal{X} \times \mathcal{X} \to \mathbb{R}$ which associates to every input pair $(\boldsymbol{x}, \boldsymbol{x}') \in \mathcal{X} \times \mathcal{X}$ a covariance parameter $k(\boldsymbol{x}, \boldsymbol{x}') = \mathbb{E}[(f(\boldsymbol{x}) - \mu(\boldsymbol{x}))(f(\boldsymbol{x}') - \mu(\boldsymbol{x}'))]$. We write $\mathsf{f} \sim \mathcal{GP}(\mu, k)$. Given data $\boldsymbol{X} = [\boldsymbol{x}_1^\top; \ldots; \boldsymbol{x}_N^\top]^\top \in \mathbb{R}^{N \times d}$ and corresponding observations $\boldsymbol{y} = [y_1, \ldots, y_N]^\top \in \mathbb{R}^N$, the posterior mean and covariance of $\mathsf{f}$ are given by[1]

$$\mu_N(\boldsymbol{x}) = \mu(\boldsymbol{x}) + \boldsymbol{k}(\boldsymbol{x}, \boldsymbol{X})(\boldsymbol{K}_N + \epsilon^2 \boldsymbol{I}_N)^{-1}(\boldsymbol{y} - \mu(\boldsymbol{X})), \qquad \text{and} \tag{3}$$

$$k_N(\boldsymbol{x}, \boldsymbol{x}') = k(\boldsymbol{x}, \boldsymbol{x}') - \boldsymbol{k}(\boldsymbol{x}, \boldsymbol{X})(\boldsymbol{K}_N + \epsilon^2 \boldsymbol{I}_N)^{-1}\boldsymbol{k}(\boldsymbol{X}, \boldsymbol{x}'). \tag{4}$$

GPs may be constructed via the weight-space view or the function-space view (Rasmussen & Williams, 2006, chapter 2.1, 2.2). We adopt a weight-space view in order to efficiently generate GP samples with a functional form.

### 2.2   Thompson sampling for Bayesian Optimisation

We outline the generic Thompson sampling procedure in Algorithm 3, where we use a GP sample $\mathsf{f}_t$ in place of the acquisition function $\alpha_t$ in (2). Of particular interest are the key steps involving drawing a sample from the GP (line 7) and identifying its maximum (line 8).

**Sampling the posterior.** Although we may naively and exactly sample from the posterior, this approach presents two challenges. Firstly, with each new observation point, we must resolve (3) and (4) for the updated test points. This process incurs a computational cost of $\mathcal{O}(N^3)$, where $N$ is the number of test points. Secondly, while the functional forms of the mean and covariance are available, the sample itself lacks an easily evaluable or differentiable functional form. To address these challenges, the prevailing strategy in Thompson sampling involves drawing a posterior sample using random features (Rahimi & Recht, 2007; Wilson et al., 2020). This framework, while providing a statistically approximate sample, effectively circumvents the

---

[1]Since $\boldsymbol{K}_N$ is constructed iteratively at each round of BO, in practice we may use block Cholesky (Golub & van Loan, 2013, §4.2.9) to compute (3) and (4) in $\mathcal{O}(N^2)$ time per round. This is not for free of course, as we will still accumulate $\mathcal{O}(N^3)$ time over the course of $N$ rounds.

---

**Algorithm 1** Difference-of-convex algorithm

---

1: **Input:** Convex functions $g_1, g_2$ such that $g(\cdot) = g_1(\cdot) - g_2(\cdot)$, initial point $\boldsymbol{x}_0 \in \mathcal{X}$, max iterations $T$, other stopping criteria (e.g. gradient tolerance)
2: **for** $t = 0$ **to** $T - 1$ **do**
3:     **if** stopping criteria met **then**
4:         **break**
5:     Let $h(\boldsymbol{x}) = g_1(\boldsymbol{x}) - \big(g_2(\boldsymbol{x}_t) + \langle \nabla g_2(\boldsymbol{x}_t), \boldsymbol{x} - \boldsymbol{x}_t \rangle\big)$
6:     $\boldsymbol{x}_{t+1} \leftarrow \arg\min_{\boldsymbol{x} \in \mathcal{X}} h(\boldsymbol{x})$. As $h$ is convex, can solve for $\boldsymbol{x}_{t+1}$ using any local optimiser such as L-BFGS.

---

aforementioned issues. Specifically, it yields a functional form for the sample in $\mathcal{O}(M^3)$ time, where $M$ is the number of random features. This cost is front-loaded, as the sample is generated only once and can subsequently be evaluated at any number of test points without requiring re-computation.

**Optimising the posterior sample: global vs. local optima.** Generating a posterior sample as above will yield a continuous, differentiable and generally non-convex function. Thus, we may attempt to optimise it using combinatorial search methods, gradient-based methods, or combinations of both. Importantly, we note that typical Bayesian optimisation regret bounds Srinivas et al. (2010); Chowdhury & Gopalan (2017) require that the global maximum of the acquisition function, or Thompson sample, is found exactly at each round. In practice, this can rarely be guaranteed. Rather, a local optimisation method like L-BFGS is usually multi-started across the domain and the best point is returned Balandat et al. (2020). This tension between theory and practice has been acknowledged in the literature Kim & Choi (2019). In this paper, we introduce a method to deconstruct Thompson samples such that they can be (locally) optimised with the difference-of-convex algorithm, rather than L-BFGS, and compare these two approaches. Theory exploring how inexact acquisition solutions affect the final regret bound are beyond the scope of this paper, although an important research direction.

### 2.3 Difference-of-convex algorithm

A function $g(\boldsymbol{x})$ is categorised as *difference of convex* (DC) if it can be expressed as

$$g(\boldsymbol{x}) = g_1(\boldsymbol{x}) - g_2(\boldsymbol{x}), \tag{5}$$

where $g_1$ and $g_2$ are convex and possibly non-smooth, and $\boldsymbol{x}$ is over a bounded compact domain (Le Thi & Tao, 2005). We refer to the right hand side of (5) as a *DC decomposition* of $g$. DC functions are non-convex in general, however they possess desirable structure that makes them more amenable to optimisation than non-DC functions. Particularly, the difference-of-convex algorithm (DCA) may be utilised (Algorithm 1). For minimisation, DCA will take some point $\boldsymbol{x}_t$, form a convex approximation about $\boldsymbol{x}_t$ using $g_1$ and $g_2$, and then solve this convex problem using some local optimiser such as L-BFGS to find $\boldsymbol{x}_{t+1}$. This may be likened to a vanilla Newton method, where successive quadratic approximations are used. The difference is that rather than using a quadratic approximation, we may use a convex approximation formed using $g$ itself. It is important to note that, as with L-BFGS, DCA can only promise local stationary points of the objective. Any $L$-smooth function can be represented as DC, although with some practicality caveats (see Section C).

## 3 Difference-of-convex Thompson sampling

We now introduce our proposed framework for efficiently generating approximate GP posterior samples with a functional form that facilitates difference-of-convex optimisation. Approximate samples are drawn from the posterior by combining elements from the approximate random-feature model with the exact Bayesian update. We do so by additively decomposing samples from the posterior into a prior and posterior contribution. In doing so, we retain the ability to optimise an explicit functional form, with approximations only introduced in the prior contribution, and favourable $\mathcal{O}(N^3)$ computational cost for $N$ data points (as opposed to test points).

### 3.1 Sampling GP prior

To generate a GP sample with an explicit functional form, we adopt the weight-space view, employing the popular approach of random features (Neal, 1995; Rahimi & Recht, 2007).

**Random feature approximations.** Let $\boldsymbol{\psi}^{(M)} : \mathcal{X} \to \mathbb{R}^M$ be an arbitrary feature mapping. We observe that for $\boldsymbol{\beta} \sim \mathcal{N}(\mathbf{0}_M, \eta_\beta^2 M^{-1} \boldsymbol{I}_M)$, the random function

$$\mathsf{f}(\cdot) = \boldsymbol{\beta}^\top \boldsymbol{\psi}^{(M)}(\cdot), \tag{6}$$

is a GP with the mean function $\mu(\boldsymbol{x}) \triangleq \mathbb{E}[\mathsf{f}(\boldsymbol{x})] = 0$, and a covariance function, given by

$$\hat{k}(\boldsymbol{x}, \boldsymbol{x}') \triangleq \mathbb{E}\big[\mathsf{f}(\boldsymbol{x})\mathsf{f}(\boldsymbol{x}')\big] = \sum_{i=1}^M \mathbb{E}\big[\beta_i^2 \psi_i^{(M)}(\boldsymbol{x})\psi_i^{(M)}(\boldsymbol{x}')\big] = \frac{\eta_\beta^2}{M} \boldsymbol{\psi}^{(M)}(\boldsymbol{x})^\top \boldsymbol{\psi}^{(M)}(\boldsymbol{x}'). \tag{7}$$

Informally, such a finite feature model approximates a GP with a kernel $k$ given by a limiting kernel. That is,

$$k(\boldsymbol{x}, \boldsymbol{x}') \triangleq \lim_{M \to \infty} \frac{\eta_\beta^2}{M} \boldsymbol{\psi}^{(M)}(\boldsymbol{x})^\top \boldsymbol{\psi}^{(M)}(\boldsymbol{x}'), \tag{8}$$

if such a limit exists. Next, we consider the two special cases of random Fourier features and random neural network features. Consider the $i$th feature mapping as

$$\psi_i^{(M)}(\boldsymbol{x}) = \varphi(\mathbf{w}_i^\top \boldsymbol{x}), \tag{9}$$

for some function $\varphi$ (applied element-wise) and random i.i.d. vector $\mathbf{w}_i$. By (8), this yields, for all $i$,

$$k(\boldsymbol{x}, \boldsymbol{x}') = \eta_\beta^2 \mathbb{E}\big[\varphi(\mathbf{w}_i^\top \boldsymbol{x})\varphi(\mathbf{w}_i^\top \boldsymbol{x}')\big], \tag{10}$$

by the law of large numbers.

**Random Fourier features.** Bochner's theorem (Rasmussen & Williams, 2006, § 4.2.1) states that any stationary kernel $k(\boldsymbol{x} - \boldsymbol{x}')$ may be represented as the Fourier transform of a probability measure, and vice versa. For example, when $\varphi$ is a complex exponential function and $\mathbf{w}_i \sim \mathcal{N}(\mathbf{0}_M, \eta_w^2 \boldsymbol{I}_M)$, the kernel (10) is the squared exponential. This corresponds to a Fourier transform of the probability measure of $\mathbf{w}$, and a Monte Carlo estimate of this is often referred to as the random Fourier feature model (Rahimi & Recht, 2007).

**Random neural network features.** More generally, $\varphi$ may be arbitrary, which means (10) will not necessarily produce stationary GP kernels. We consider a finite random feature model with $M$ features, in which each $\mathbf{w}_i$ is sampled once and then held fixed during posterior updates. Using (6) and (9), we consider the model

$$\mathsf{f}(\boldsymbol{x}) = \boldsymbol{\beta}^\top \varphi(\mathbf{W}\boldsymbol{x}),$$

where $\varphi$ applies element-wise and $\mathbf{W} = [\mathbf{w}_1, \ldots, \mathbf{w}_M]^\top \in \mathbb{R}^{M \times d}$ is a random weight matrix. Observe that $\mathsf{f}(\boldsymbol{x})$ resembles a single hidden-layer neural network with an activation function $\varphi$, randomly sampled (and then fixed) hidden layer weights $\mathbf{W}$ (Neal, 1995; Williams, 1997), and Bayesian updating of output layer weights $\boldsymbol{\beta}$. Importantly, note that for an arbitrary choice of $\varphi$, we may approximate the respective kernel via (7) Rahimi & Recht (2007). However, for some choices of $\varphi$ we may derive the exact kernel analytically. One case is taking $\varphi$ to be the ReLU function. As $M \to \infty$ we obtain the first order arc-cosine kernel (Cho & Saul, 2009), given by

$$k(\boldsymbol{x}, \boldsymbol{x}') = \frac{\eta_\beta^2 \eta_w^2}{2\pi} \|\boldsymbol{x}\|\|\boldsymbol{x}'\|\big(\sin\theta + (\pi - \theta)\cos\theta\big), \tag{11}$$

---

**Algorithm 2** Posterior samples with random features and pathwise updates

---

1: **Input:** dataset $\mathcal{D}_{t-1}$, feature mapping $\varphi(\cdot)$, corresponding exact kernel $k(\cdot, \cdot)$, number of random features $M$. Draw, and fix, $[W]_{i,j} \sim^{\text{iid}} \mathcal{N}(0, \eta_w^2)$.
2: Draw $\boldsymbol{\beta} \sim \mathcal{N}\left(\mathbf{0}_M, \frac{\eta_\beta^2}{M} \boldsymbol{I}_M\right)$
3: Define prior sample $\mathsf{f}_0(\boldsymbol{x}) = \boldsymbol{\beta}^\top \varphi(\boldsymbol{W}\boldsymbol{x})$
4: Compute the posterior $\mathsf{f}_t(\boldsymbol{x})$ via (13)

---

where $\theta = \cos^{-1} \boldsymbol{x}^\top \boldsymbol{x}'/(\|\boldsymbol{x}\|\|\boldsymbol{x}'\|)$ is the angle between $\boldsymbol{x}$ and $\boldsymbol{x}'$. These random ReLU features and their corresponding arc-cosine kernel will be used later in the experiments section. A similar kernel may be derived for leaky ReLU (Tsuchida et al., 2018).

Now, consider $\boldsymbol{\beta} \sim \mathcal{N}(\mathbf{0}_M, M^{-1}\eta_\beta^2 \boldsymbol{I}_M)$ and $\boldsymbol{W} \in \mathbb{R}^{M \times d}$ such that $W_{i,j} \sim \mathcal{N}(0, \eta_w^2)$. Let

$$\mathsf{f}_0(\boldsymbol{x}) = \boldsymbol{\beta}^\top \varphi(\mathbf{W}\boldsymbol{x}) \tag{12}$$

be the approximate prior sample from a GP with zero mean and kernel $k$, where $k$ is the equivalent kernel corresponding to the feature mapping $\varphi$. Here, $\eta_\beta$ and $\eta_w$ are standard deviations for each $\boldsymbol{\beta}$ and $\mathsf{w}$ respectively, and are treated as tuneable hyperparameters[2], although theoretical results require defining $\eta_\beta$ to be increasing with each BO iteration (see Section B).

### 3.2 Prior to posterior via pathwise updates

Given a prior sample generated with (12), we may use pathwise updates (Wilson et al., 2021) to obtain a corresponding posterior sample path $\mathsf{f}_t$, given by

$$\underbrace{\mathsf{f}_t(\boldsymbol{x})}_{\text{posterior}} = \underbrace{\mathsf{f}_0(\boldsymbol{x})}_{\text{prior}} + \underbrace{\boldsymbol{k}(\boldsymbol{x}, \boldsymbol{X})\big(\boldsymbol{K}_N + \epsilon^2 \boldsymbol{I}_N\big)^{-1}\big(\boldsymbol{y} - \mathsf{f}_0(\boldsymbol{X}) - \boldsymbol{\epsilon}\big)}_{\text{update}}, \tag{13}$$

where we define $\mathbf{f}_0(\boldsymbol{X}) = [\mathsf{f}_0(\boldsymbol{x}_1), \ldots, \mathsf{f}_0(\boldsymbol{x}_N)]^\top$ and $\boldsymbol{\epsilon} \sim \mathcal{N}(\mathbf{0}_N, \epsilon^2 \boldsymbol{I}_N)$. This functional form is well-suited for optimising the sample, as the linear solve is performed only once, rather than for each test point. Unlike the functional form obtained through Bayesian regression on $\boldsymbol{\beta}$ in feature space (Rahimi & Recht, 2007; Snoek et al., 2015), this approach ensures that the posterior mean and mode precisely match the true process. It also significantly simplifies the theoretical analysis by eliminating the need to account for an inexact posterior mean. We summarise this posterior sampling method in Algorithm 2.

**Efficiency.** This method of drawing a posterior sample enjoys scaling as $\mathcal{O}(N^3)$ for $N$ data points. This scaling is well-suited to Bayesian optimisation settings, where the number of data points is typically small due to the high cost of evaluating $f$. In contrast, directly sampling from the posterior without using pathwise updates scales as $\mathcal{O}(M^3)$ for $M$ random features.

**Theoretical results.** Algorithm 3 with sample paths generated using Algorithm 2 achieves sublinear cumulative regret, as formalised in Theorem 3.1. The proof, provided in Section B, follows a strategy similar to that of Chowdhury & Gopalan (2017), Mutny & Krause (2018), Dai et al. (2020) and Vakili et al. (2020). However, our approach simplifies the proof by leveraging the combination of random features and pathwise updates, ensuring an exact (rather than approximate) posterior mean.

**Theorem 3.1.** *Suppose* $\max_{\boldsymbol{x} \in \mathcal{X}} f(\boldsymbol{x}) \leq B < \infty$, *and* (2) *with* $\alpha_t(\boldsymbol{x}) = \mathsf{f}_t(\boldsymbol{x})$ *is solved exactly. After* $T$ *rounds, the cumulative regret of Algorithm 3 with sample paths generated using Algorithm 2 can be expressed as*

$$R_T = \tilde{\mathcal{O}}\bigg((B+1)\gamma_T\sqrt{T}\bigg),$$

---

[2]In practice, we also append $\boldsymbol{x}$ with a constant bias such that $\boldsymbol{x} \leftarrow [\boldsymbol{x}^\top, \eta_b/\eta_w]^\top$. This allows the bias variance to be tuned independently via $\eta_b^2$.

---

**Algorithm 3** Difference-of-convex Thompson sampling

---
1: **Input:** domain $\mathcal{X}$, noisy objective function $f$, number of initial observations $N$, number of BO iterations $T$
2: Generate $N$ input points $\boldsymbol{x}_i^{\text{init}}$ and corresponding output points $y_i^{\text{init}} \approx f(\boldsymbol{x}_i^{\text{init}})$ for $i = 1, \ldots, N$
3: $\mathcal{D}_0 \leftarrow \{(\boldsymbol{x}_i^{\text{init}}, y_i^{\text{init}})\}_{i=1}^N$
4: $(\boldsymbol{x}_{\max}, y_{\max}) \leftarrow \max\{y_i^{\text{init}}, i = 1, \ldots, N\}$
5: **for** $t = 1$ **to** $T$ **do**
6:      Build a GP posterior $p\big(f_t(\boldsymbol{x})|\mathcal{D}_{t-1}\big)$
7:      Generate a difference-of-convex (DC) posterior sample path $\mathsf{f}_t(\boldsymbol{x}) \sim p\big(f_t(\boldsymbol{x})|\mathcal{D}_{t-1}\big)$ via Algorithm 2
8:      Find $\boldsymbol{x}_t$ by solving (2) for $\alpha_t(\boldsymbol{x}) = \mathsf{f}_t(\boldsymbol{x})$ via Algorithm 1
9:      $y_t \approx f(\boldsymbol{x}_t)$
10:     $\mathcal{D}_t \leftarrow \mathcal{D}_{t-1} \cup \{(\boldsymbol{x}_t, y_t)\}$
11:     $(\boldsymbol{x}_{\max}, y_{\max}) \leftarrow \max(y_{\max}, y_t)$
12: **return** $\{\boldsymbol{x}_{\max}, y_{\max}\}$

---

where $\gamma_T$ is the maximum information gain after $T$ rounds, and $\tilde{\mathcal{O}}(\cdot)$ denotes asymptotic order, ignoring log factors.

This result is unsurprising and is similar to regret bounds for existing Thompson sampling methods. The key difference is that the mean of our approximate sample $\mathsf{f}_t$ exactly matches that of the true GP, with only the variance being an approximation. We show that this approximate variance $\hat{\sigma}_t(\boldsymbol{x})^2$ is given by

$$\hat{\sigma}_t(\boldsymbol{x})^2 = \frac{\eta_\beta^2}{M}\varphi(\boldsymbol{W}\boldsymbol{x})^\top \varphi(\boldsymbol{W}\boldsymbol{x}) - \frac{\eta_\beta^2}{M}\varphi(\boldsymbol{W}\boldsymbol{x})^\top \varphi(\boldsymbol{X}\boldsymbol{W}^\top)^\top \boldsymbol{A}k(\boldsymbol{X},\boldsymbol{x}) + \zeta(\boldsymbol{x}), \tag{14}$$

where

$$\zeta(\boldsymbol{x}) = -\frac{\eta_\beta^2}{M}k(\boldsymbol{x},\boldsymbol{X})\boldsymbol{A}\varphi(\boldsymbol{X}\boldsymbol{W}^\top)\varphi(\boldsymbol{W}\boldsymbol{x}) + k(\boldsymbol{x},\boldsymbol{X})\boldsymbol{A}\Big(\frac{\eta_\beta^2}{M}\varphi(\boldsymbol{X}\boldsymbol{W}^\top)\varphi(\boldsymbol{W}\boldsymbol{X}^\top) + \epsilon^2\boldsymbol{I}_N\Big)\boldsymbol{A}k(\boldsymbol{X},\boldsymbol{x})$$

and $\boldsymbol{A} = \big(k(\boldsymbol{X},\boldsymbol{X}) + \epsilon^2\boldsymbol{I}_N\big)^{-1}$. Although this expression may seem unwieldy, our algorithm does not require its explicit evaluation. As $M \to \infty$, the terms involving $1/M$ approach the true kernel expressions via (8), leading to $\zeta(\boldsymbol{x}) \to 0$. This ensures that (14) matches the posterior variance of the true GP in (4).

Theoretical regret bounds on BO algorithms generally require that the acquisition function, or Thompson sample, be maximised globally and exactly, which is difficult to guarantee in practice. In existing literature, this subproblem is usually solved locally and inexactly, for example a local maximiser of $\alpha(\boldsymbol{x})$ may be considered "good enough". While our method also only guarantees local solutions, we demonstrate how constructing a GP posterior sample using the above method can make this subproblem easier.

## 3.3 Posterior sample optimisation using DCA

Recall the definition of difference-of-convex (DC) and the DC algorithm in Section 2.3. Observe that by using convex activations (such as ReLUs) in a shallow neural network, we admit a DC decomposition of the network, and thus we may find approximate DC decompositions of arbitrary functions (Awasthi et al., 2024).

**DC decomposition of posterior sample.** We may express (13) as

$$\mathsf{f}_t(\boldsymbol{x}) = \sum_{i=1}^M \beta_i \varphi(\boldsymbol{w}_i^\top \boldsymbol{x}) + \sum_{j=1}^N \mathsf{a}_j k(\boldsymbol{x}, \boldsymbol{X}_{j,:}), \tag{15}$$

where $\mathbf{a} = [\mathsf{a}_1, \ldots, \mathsf{a}_N]^\top = \big(\boldsymbol{K}_N + \epsilon^2\boldsymbol{I}_N\big)^{-1}\big(\boldsymbol{y} - \mathbf{f}_0(\boldsymbol{X})\big)$. This representation allows us to interpret the posterior as a linear combination of basis functions.

|  | ReLU (with DC, **ours**) | ReLU (no DC) | Sq. exp. | EI | UCB |
|---|---|---|---|---|---|
| Granular sim | $1.37 \pm 1.23$ | $3.15 \pm 2.36$ | $8.88 \pm 3.98$ | $6.32 \pm 1.49$ | $5.01 \pm 3.12$ |
| Rosenbrock | $2.19 \pm 0.37$ | $2.45 \pm 0.47$ | $3.67 \pm 0.35$ | $11.79 \pm 7.61$ | $4.14 \pm 0.63$ |
| Michaelwicz | $5.38 \pm 0.19$ | $5.50 \pm 0.21$ | $8.01 \pm 0.10$ | $5.67 \pm 0.34$ | $5.97 \pm 0.19$ |
| Rastrigin | $48.65 \pm 3.82$ | $61.71 \pm 5.07$ | $66.75 \pm 4.51$ | $109.31 \pm 7.76$ | $92.90 \pm 4.97$ |
| Synthetic | $5.62 \pm 3.85$ | $8.77 \pm 3.55$ | $7.81 \pm 3.51$ | $67.58 \pm 2.85$ | $7.10 \pm 3.68$ |
| NN tuning | $11.09 \pm 0.43$ | $11.76 \pm 0.70$ | $13.77 \pm 0.56$ | $12.06 \pm 0.73$ | $12.14 \pm 1.33$ |

Table 1: BO for minimisation. Mean and 95% confidence interval of lowest function value found, over several trial runs. For details, see Section A.

Choosing convex nonnegative feature mappings $\varphi$, e.g., ReLUs, leads to a convex equivalent kernel $k$. To see this, write (10) as

$$k(\boldsymbol{x}, \boldsymbol{x}') = \int_{\Omega} \varphi(\boldsymbol{w}^{\top}\boldsymbol{x})\varphi(\boldsymbol{w}^{\top}\boldsymbol{x}')d\mu(\boldsymbol{w}),$$

where $\mu$ is a nonnegative probability measure over the sample space $\Omega$. Then $k$ is also a convex function with respect to each argument, while keeping the other fixed. Consequently, for any convex feature mapping, all basis functions in (15) are convex, allowing to construct a DC decomposition of $f_t$ as follows.

For each $\beta_i, a_j \in \mathbb{R}$ and $i = 1, \ldots, M$ and $j = 1, \ldots, N$, we can mask $\boldsymbol{\beta}$ and $\mathbf{a}$ into positive and negative components as

$$\beta_i^+ := \max\{0, \beta_i\}, \quad \beta_i^- := \min\{0, \beta_i\},$$
$$a_j^+ := \max\{0, a_j\}, \quad a_j^- := \min\{0, a_j\},$$

with each component stacking into their corresponding vectors $\boldsymbol{\beta}^+, \boldsymbol{\beta}^- \in \mathbb{R}^M$ and $\boldsymbol{a}^+, \boldsymbol{a}^- \in \mathbb{R}^N$. We may then express (15) as

$$f_t(\boldsymbol{x}) = \underbrace{\sum_{i=1}^{M} \beta_i^+ \varphi(\boldsymbol{w}_i^{\top}\boldsymbol{x}) + \sum_{j=1}^{N} a_j^+ k(\boldsymbol{x}, \boldsymbol{X}_{j,:})}_{\text{convex}} - \underbrace{\sum_{i=1}^{M} -\beta_i^- \varphi(\boldsymbol{w}_i^{\top}\boldsymbol{x}) + \sum_{j=1}^{N} -a_j^- k(\boldsymbol{x}, \boldsymbol{X}_{j,:})}_{\text{convex}},$$

which provides a DC decomposition of $f_t$ as in (5).

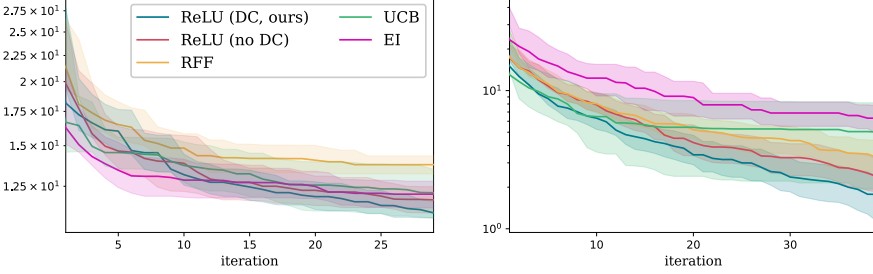

Figure 2: **Left:** Bayesian optimisation for neural network hyperparameter tuning, minimising test loss across 9 hyperparameters. **Right:** Calibrating an expensive granular simulation over 10 design variables to minimise squared distance to a target *angle of repose* of 26°, with each objective function call taking approximately 20 minutes to compute on a HPC cluster. Each feature type is run over 30 trials. Solid line denotes mean, shading is 95% confidence.

**Difference of convex algorithm.** Having a DC decomposition of $f_t$ enables optimisation using the Difference of Convex Algorithm (DCA), which guarantees convergence to local stationary points (Le Thi &

Tao, 2005; Le Thi & Pham Dinh, 2018). DCA iteratively approximates local optima by solving a series of convex subproblems (Algorithm 1). Each convex subproblem can be addressed using any standard convex optimisation method, such as gradient descent or L-BFGS.

## 4 Experiments

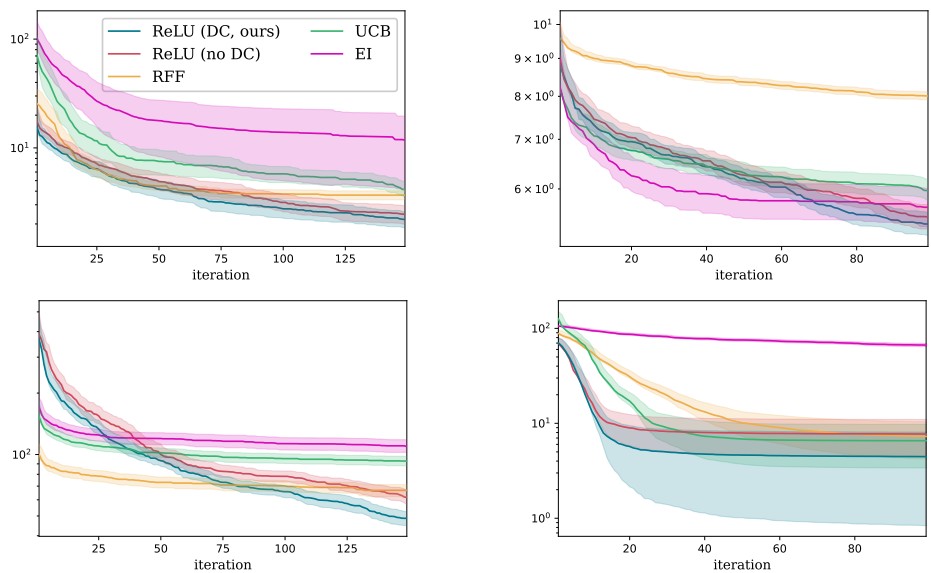

Figure 3: Benchmark functions for minimisation, 50 trials each. Solid line indicates the mean, shading indicating 95% confidence interval. Vertical axis is best point found so far. **Top left:** 6D Rosenbrock function. **Top right:** 10D Michaelwicz function. **Bottom left:** 10D Rastrigin function. **Bottom right:** 10D synthetic function generated as a GP sample via (12), with random ReLU features. Further details in Section A.

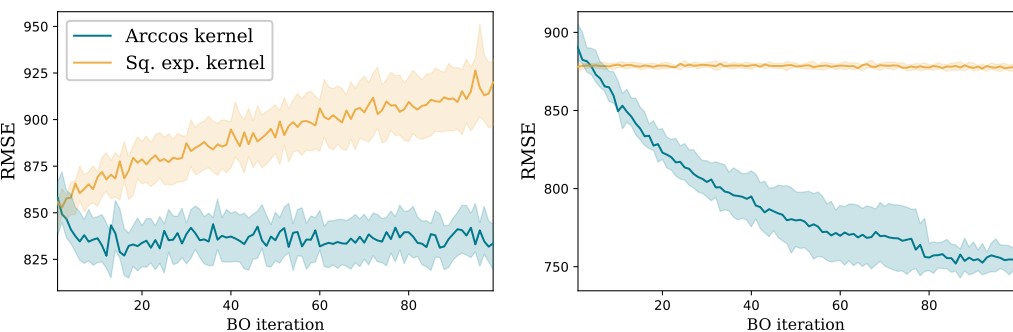

Figure 4: Goodness-of-fit. Average RMSE of objective function vs. GP surrogate of 100 randomly sampled points at each BO iteration on a 6D Rosenbrock function (left) and a 10D Rastrigin function (right), over 50 trials.

We run DCTS with where sample paths are generated using random ReLU features and the exact limiting arc-cosine kernel (equation (11)). Due to the convexity of the random features, we may optimise the resulting sample paths using DCA (Algorithm 1). This is compared to the same features/kernel setup where we ignore the DC structure and simply optimise samples using L-BFGS. These are also compared to generic Thompson sampling using a squared exponential kernel and random Fourier features (RFF), expected improvement (EI) and upper confidence bound (UCB) using squared exponential kernels. Results are summarised in Table 1, with convergence plots in Figure 2 and Figure 3. Further experimental details are given in Section A.

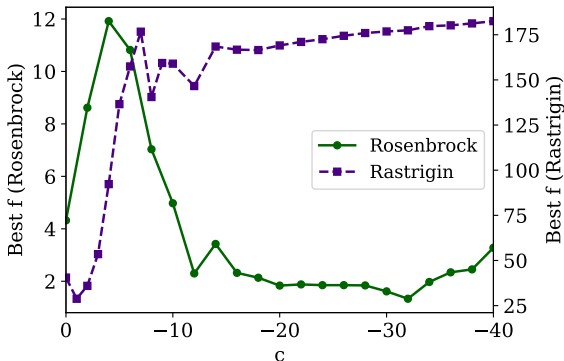

Figure 5: Best objective function point for minimisation after 100 iterations for various values of $c$ in equation (16), lower is better. Parameters of each objective function are per Section A. While Rastrigin performs best with shallow quadratic bowls, Rosenbrock performs best with steeper bowls.

**Hyperparameters and prior mean.** While ReLU networks are indeed universal approximators, samples generated with random ReLU features may grow large away from observed data points. This "boundary issue" is discussed in Swersky (2017). The resulting effect is that the acquisition optimiser may be coaxed towards edges of the domain where the sample has grown large, especially in high dimensions, which can lead to inefficient exploration. While this is not necessarily unreasonable given no prior knowledge of the objective function, for most BO problems we intuit that the maximum lies somewhere away from the boundaries of our domain. Thus, in order to coax the optimiser away from the boundary, it can be useful to use a quadratic "concave bowl" prior mean given by

$$\mu_0(\boldsymbol{x}) = c\|\boldsymbol{x} - \boldsymbol{m}\|_2^2 \tag{16}$$

where $\boldsymbol{m} \in \mathbb{R}^d$ is the midpoint of the domain and $c \in (-\infty, 0]$. Here, we select various values of $c$ per problem. We also find that the convergence speed of the routine is sensitive to $c$, like many hyperparameters in BO. Figure 5 shows DCTS performance for various values of $c$.

Similarly, although theory suggests $\beta_t$ must increase, in practice we may get better results by using a fixed value. We use various parameter values for our experiments, detailed in Section A.

**Design optimisation and granular simulations.** Bayesian optimisation is frequently applied to design optimisation problems in science and engineering, which are often black-box and computationally expensive (Kennedy & O'Hagan, 2001). Here, we focus on a real-world application involving a granular material simulation within a rotating drum, which is a scenario prevalent in industrial settings and a critical tool for calibrating material properties to match with observed behaviours. Our simulation utilises the discrete element method (DEM), which is a state of the art approach widely employed for its effectiveness in modelling granular flows within industrial machinery (Cleary, 2009). Beyond the specific case considered here, DEM is also integral to research efforts in advanced manufacturing, including metal 3D printing (Phua et al., 2021) and optimising resource intensive processes such as energy efficient particle breakage in minerals processing applications (Delaney et al., 2015).

We consider the commonly encountered problem of calibration of an expensive DEM granular material flow simulation to match an experimentally measured *angle of repose* (Mead et al., 2012). We seek to minimise the squared difference between the simulated and target angle given 10 input parameters, with results given in Figure 2.

**NN hyperparameter tuning.** Machine learning hyperparameter optimisation is a typical use-case of Bayesian optimisation (Klein et al., 2017; Chen et al., 2022). We train a fully-connected neural network with two hidden layers on the MNIST dataset, optimising 9 hyperparameters; see Section A for details.

We also experiment on 4 synthetic benchmark functions, shown in Figure 3 and described in Section A.

### 4.1 Discussion and limitations

The success of any BO algorithm generally relies on two factors: the choice of kernel and kernel hyperparameters, and the ability to accurately optimise the acquisition function. Our work focuses on the latter, showing that if the user is willing to restrict their kernel choice (as described), then optimisation of the acquisition function will come easier. It is common wisdom to assume that Matérn kernels are generally most performant for BO. Figure 4 shows Thompson sampling where at each iteration, 100 points of the sample within the domain are uniformly randomly sampled, and their RMSE against the objective function plotted, providing a goodness-of-fit metric for the surrogate. In these example cases, the GP with the arccos kernel (constructed from ReLU features) provides a closer fit to the objective, suggesting this common wisdom may not be entirely accurate.

Our experiments suggest that exploiting DC structure during acquisition optimisation gives better results than not, all else being the same. However, BO notoriously exhibits high hyperparameter sensitivity (Wang & Freitas, 2014; Berkenkamp et al., 2019), and the ideal choice of kernel for a particular objective function is rarely obvious beforehand. The main limitation of our method is that the features used to construct the kernel *must* be convex, which places restrictions on the choice of kernel. (Note, it is still assumed that the objective function is generally non-convex.) We also find that optimising the GP hyperparameters using log-likelihood maximisation to a achieve a well-fitting surrogate does not necessarily improve overall BO performance. That said, if the features are non-convex but have a bounded second derivative, we may find a somewhat artificial DC decomposition (see Section C), although with poor results, suggesting our method for finding a "natural" DC decomposition is more performant.

## 5 Conclusion

Although Bayesian optimisation is a powerful black-box optimisation technique, it relies on accurate optimisation of either an acquisition function or a Thompson sample at each round, and theoretical results are reliant on this optimisation being done exactly and globally. While this may be tractable in low dimensions, difficulty arises in higher dimensions as the search hypervolume becomes large. To mitigate this intractability, we develop a Thompson sampling method such that, under certain kernels, the sample drawn possesses a desirable *difference-of-convex* structure, which allows it to be optimised more efficiently using a DC algorithm. While theoretical connections between Gaussian processes and neural networks are well established, we wish for future work to explore DC structures more broadly in the context of machine learning.

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

## A  Experiment details

For all experiments, our Bayesian optimisation routine begins as follows:

- Draw a GP prior sample with $M = 1000$ random features via (12), for some feature variance $\beta_0$ and lengthscale $l$ (if appropriate). We take a quadratic bowl prior mean given by (16) for some $c$.

- Select $N_{\text{init}}$ random points within the domain $\mathcal{X}$ via Latin hypercube sampling, and sample the objective function $f$ at these points to obtain an initial dataset $\mathcal{D} = (\boldsymbol{X}, \boldsymbol{y})$.

- Condition the GP on $\mathcal{D}$ with pathwise updates using the corresponding exact limiting kernel via (13).

- Use log likelihood maximisation (Rasmussen & Williams, 2006, §5.4) with an Adam optimiser to find a reasonable set of hyperparameters for the GP surrogate (variance $\beta_0$ and lengthscale $l$).

From here, the main BO loop is as follows:

1. Draw a GP posterior sample via (12) and (13). We may scale sample variance at round $t$ as $\beta_t = \beta_0 \log(t+1)$ to encourage exploration. Theoretical results are often reliant on this increasing $\beta_t$ for convergence, but we find varying results in practice: sometimes it will aid optimisation, sometimes it will hinder it.

2. Use DIRECT Jones et al. (1993) to find an initial point at which to run the local optimiser. We use the following parameters: $10^{-9}$ function tolerance, $10^3 d$ maximum number of function evaluations, $10^4 d$ maximum number of iterations, and $10^4 d$ maximum number of rectangle divisions.

3. Initialise a local optimiser at the point found above. When using ReLU random features with the corresponding arc-cosine kernel, we exploit the DC structure and optimise via Algorithm 1 with a gradient break tolerance of $10^{-8}$ for the outer loop, and $10^{-2}$ for the inner loop. We cap each inner loop at 10 iterations, the outer loop at 100 iterations. For all others, we use L-BFGS with a gradient break tolerance of $10^{-8}$, capped at 1000 iterations. We also contrast ReLU features with DCA against ReLU features with L-BFGS.

4. Observe the objective function at the point found by the local optimiser, and add to the dataset $\mathcal{D}$.

We test Thompson sampling using ReLU features and the corresponding arc-cosine kernel (see (11)) with sample optimisation via DCA (Algorithm 1), ReLU features with sample optimisation via L-BFGS, Thompson sampling with the squared exponential kernel, expected improvement with a squared exponential kernel, and upper confidence bound using a squared exponential kernel Srinivas et al. (2010).

**Benchmark functions.**  We test empirical performance on 3 benchmark functions for minimisation: the 6D Rosenbrock function, given by

$$f(\boldsymbol{x}) = -\sum_{i=1}^{5} \left[ a(x_{i+1} - x_i^2)^2 + (b - x_i)^2 \right],$$

for $a = 1$, $b = 1$, $\boldsymbol{x} \in [-5, 5]^6$, $N_{\text{init}} = 18$ and quadratic mean $c = -35$, with $\beta_t$ increasing. A 10D Michaelwicz function, given by

$$f(\boldsymbol{x}) = \sum_{i=1}^{20} \sin(x_i) \sin^{2a}\left( \frac{i x_i^2}{\pi} \right) + 20,$$

for $a = 1$, $\boldsymbol{x} \in [-\pi, \pi]^{20}$, $N_{\text{init}} = 30$ and $c = -1$, with $\beta_0$ constant. The 10D Rastrigin function, given by

$$f(\boldsymbol{x}) = 10a + \sum_{i=1}^{10} \left[ x_i^2 - a\cos(2\pi x_i) \right],$$

for $a = 10$, $\boldsymbol{x} \in [-10, 10]^{10}$, $N_{\text{init}} = 30$ and $c = -1$, with $\beta_0$ constant.

**Synthetic functions.** We generate synthetic functions via (12) in $d = 10$ dimensions using $M = 100$ random ReLU features, such that the objective function is of the form

$$f(\boldsymbol{x}) = \boldsymbol{\beta}^\top \varphi(\boldsymbol{W}\boldsymbol{x} + b) + 10$$

where $\varphi(\cdot)$ is the ReLU function, $\boldsymbol{W} \sim \mathcal{N}(0, 100)^{M \times d}$ and $\boldsymbol{b} \sim \mathcal{N}(0, 1)^M$.

**MNIST.** We perform 9D hyperparameter optimisation to minimise test loss over 10 trials. We train a fully-connected neural network with two hidden layers of width 64 and leaky ReLU activations with slope $\alpha \in [0.01, 0.3]$. All layers have a dropout rate $p_{\mathrm{drop}} \in [0.0, 0.5]$ and batch norm momentum $\beta_{\mathrm{b}} \in [0.1, 0.99]$. We train for 5 epochs using an Adam optimiser with a batch size of 64, $\beta_1 \in [0.8, 0.99]$, $\beta_2 \in [0.9, 0.9995]$ and learning rate decay $\lambda_{\mathrm{r}} \in [0.5, 0.99]$. We optimise the $\log_{10}$ of the learning rate $\eta$ and Adam weight decay $\lambda_{\mathrm{w}}$ such that $\log_{10}(\eta) \in [-3, 1]$ and $\log_{10}(\lambda_{\mathrm{w}}) \in [-5, -2]$ (so that the optimiser is searching over a logarithmically-scaled domain). We use gradient clipping $c \in [0.1, 5.0]$. For Bayesian optimisation, we take $N_{\mathrm{init}} = 5$, constant $\beta_0$, and $c = -80$.

**Design optimisation and granular simulations.** Design optimisation problems in science and engineering are often posed as inverse problems which involve querying computationally expensive black-box physics simulations. Forward evaluations involve specifying input design parameters such as component material, size and shape, and running a simulation around them. From here we obtain an objective function evaluation for some quantity we wish to optimise, such as drag, stability, efficiency, cost, and so on. BO may be used for calibration of the simulation parameters themselves, by treating them as input parameters in design space, and minimising the difference between simulation output and observed data. This topic is discussed in the much-cited work of Kennedy & O'Hagan (2001). We run our experiment as a 10D parameter calibration on an expensive discrete element method (DEM) granular material flow simulation, minimising the squared difference between simulated and target *angle of repose.*

At the start of the simulation, particles are generated with radii uniformly distributed about $[D_{\mathrm{min}}, D_{\mathrm{max}}]$. We take each of these as a BO input parameter such that $D_{\mathrm{min}} \in [0.07, 0.08]$ and $D_{\mathrm{max}} \in [0.081, 0.09]$. Similarly, particle XY and XZ aspect ratios are sampled uniformly from $[XY_{\mathrm{min}}, XY_{\mathrm{max}}]$ and $[XZ_{\mathrm{min}}, XZ_{\mathrm{max}}]$. We take as BO inputs $XY_{\mathrm{min}} \in [0.5, 0.8]$, $XY_{\mathrm{max}} \in [0.81, 1.0]$, $XZ_{\mathrm{min}} \in [0.5, 0.8]$ and $XZ_{\mathrm{max}} \in [0.81, 1.0]$. We take particle angularity to be sampled uniformly from $a = [a_{\mathrm{min}}, a_{\mathrm{max}}]$ where $a{\mathrm{min}} \in [2.05, 3.4]$ and $a{\mathrm{max}} \in [3.5, 5.0]$. We take coefficient of restitution $e \in [0.1, 0.5]$ and friction $\mu \in [0.5, 1.0]$. For Bayesian optimisation, we take $N_{\mathrm{init}} = 5$, constant $\beta_0$, and $c = -100$.

Each objective function evaluation requires a full simulation, which takes approximately 20 minutes.

## B  Regret analysis

Following Dai et al. (2020); Chowdhury & Gopalan (2017), we show our cumulative regret bound. Let $f(\boldsymbol{x}) : \mathcal{X} \to \mathbb{R}$, for $\mathcal{X} \subset \mathbb{R}^d$ compact and convex, be the objective function to be optimised, assumed to belong to an RKHS with associated PSD kernel $k$. Let $\mathcal{D}_{t-1}$ denote the data gathered prior to round $t$, which produce GP posterior mean and variance $\mu_{t-1}(\boldsymbol{x})$ and $\sigma_{t-1}(\boldsymbol{x})^2$ (to be used for computations during round $t$). Recalling that $\epsilon$ is the standard deviation of the observation noise, we let $\beta_t = B + \epsilon\sqrt{2(\gamma_{t-1} + 1 + \log(4/\delta))}$, for some $\delta \in (0, 1)$ and where $|f(\boldsymbol{x})|$ is bounded by $B$. Here, $\gamma_t$ is the maximum information gain on $f$ from any set of $t$ observations, defined by

$$\gamma_t = \max_{A \subset \mathcal{X}:|A|=t} I(y_A; f_A) \tag{17}$$

for some arbitrary set of points $A \subset \mathcal{X}$. This denotes the mutual information between $f_A = [f(\boldsymbol{x})]_{x \in A}$ and $y_A = f_A + \epsilon$, where $\epsilon \sim \mathcal{N}(0, \epsilon^2)$. While calculating this quantity is nontrivial, upper bounds on $\gamma_t$ for common kernels are derived in Srinivas et al. (2010, appendix C).

Let $f_t(\boldsymbol{x})$ be a GP posterior sample path generated at round $t$, sampled from $\mathcal{GP}(\mu_{t-1}(\boldsymbol{x}), \beta_t^2 \sigma_{t-1}(\boldsymbol{x})^2)$. Let $\hat{f}_t(\boldsymbol{x})$ be an approximate posterior sample taken from the same GP using random features and pathwise updates.

In some of the lemmas to follow, per Chowdhury & Gopalan (2017), at each round $t$, we restrict the decision set to be a unique discretisation $\mathcal{X}_t$ of $\mathcal{X}$, such that $|f(\boldsymbol{x}) - f([\boldsymbol{x}]_t)| \leq 1/t^2$ for all $\boldsymbol{x} \in \mathcal{X}$, where $[\boldsymbol{x}]_t$ is the closest point to $\boldsymbol{x}$ in $\mathcal{X}_t$. This is achieved by choosing a compact and convex domain $\mathcal{X} \subset [0,r]^d$ and evenly spaced discretisation $\mathcal{X}_t$ with size $|\mathcal{X}_t| = (BLrdt^2)^d$, implying that $\|\boldsymbol{x} - [\boldsymbol{x}]_t\|_1 \leq rd/(BLrdt^2) = 1/(BLt^2)$ for all $\boldsymbol{x} \in \mathcal{X}$, where $L$ is a Lipschitz constant such that

$$|f(\boldsymbol{x}) - f([\boldsymbol{x}]_t)| \leq BL\|\boldsymbol{x} - [\boldsymbol{x}]_t\|_1 \leq 1/t^2. \tag{18}$$

For technical reasons, we conduct part of the proof on the discretised grid $\mathcal{X}_t$, although our final cumulative regret bound will hold over the compact, convex domain $\mathcal{X}$.

**Lemma B.1.** *Let $\delta \in (0,1)$. For all $\boldsymbol{x} \in \mathcal{X}$, denote the event*

$$|f(\boldsymbol{x}) - \mu_{t-1}(\boldsymbol{x})| \leq \beta_t \sigma_{t-1}(\boldsymbol{x}) \tag{19}$$

*by $E^f(t)$. Then $\Pr\left(E^f(t)\right) \geq 1 - \delta/4$ for all $t \geq 1$.*

This concentrates the objective function $f$ around the posterior mean of its GP surrogate. We take theorem 2 of Chowdhury & Gopalan (2017) and an error probability of $\delta/4$.

**Lemma B.2.** *For $\boldsymbol{x} \in \mathcal{X}_t$, denote the event*

$$|f_t(\boldsymbol{x}) - \mu_{t-1}(\boldsymbol{x})| \leq \beta_t \sqrt{2\log(|\mathcal{X}_t|t^2)}\sigma_{t-1}(\boldsymbol{x}) \tag{20}$$

*by $E^{f_t}(t)$. Then $\Pr\left(E^{f_t}(t)\right) \geq 1 - 1/t^2$ for all $t \geq 1$.*

This concentrates a (true) posterior sample $f_t$ around the posterior mean and is a simpler version of Lemma 5 from Chowdhury & Gopalan (2017), which uses Lemma B4 from Hoffman et al. (2013).

*Proof.* Observe for $Z \sim \mathcal{N}(0,1)$, and any $c > 0$, that

$$\begin{aligned}
\Pr(Z > c) &= \frac{e^{-c^2/2}}{\sqrt{2\pi}} \int_c^\infty e^{(c^2-z^2)/2}\,dz \\
&= \frac{e^{-c^2/2}}{\sqrt{2\pi}} \int_c^\infty e^{-(z-c)^2/2 - c(z-c)}\,dz \\
&\leq \frac{e^{-c^2/2}}{\sqrt{2\pi}} \int_c^\infty e^{-(z-c)^2/2}\,dz \\
&= \frac{1}{2}e^{-c^2/2},
\end{aligned}$$

as $e^{-c(z-c)} \leq 1$ for $z \geq c$. By the union bound, we have that

$$\Pr(|Z| > c) \leq e^{-c^2/2},$$

and transforming by setting $Z = \left(f_t(\boldsymbol{x}) - \mu_{t-1}(\boldsymbol{x})\right)/\sigma_{t-1}(\boldsymbol{x})$ and $c = \beta_t$, and taking the union bound over all $\boldsymbol{x} \in \mathcal{X}_t$, gives

$$\Pr\left(|f_t(\boldsymbol{x}) - \mu_{t-1}(\boldsymbol{x})| \leq \beta_t \sigma_{t-1}(\boldsymbol{x})\right) \geq 1 - |\mathcal{X}_t|e^{-\beta_t^2/2}.$$

Setting $\delta = |\mathcal{X}_t|e^{-\beta_t^2/2}$, we have that

$$\Pr\left(|f_t(\boldsymbol{x}) - \mu_{t-1}(\boldsymbol{x})| \leq \sqrt{2\log(|\mathcal{X}_t|/\delta)}\sigma_{t-1}(\boldsymbol{x})\right) \geq 1 - \delta.$$

The result follows by taking $\delta = 1/t^2$. $\qquad\square$

**Lemma B.3.** *For $\boldsymbol{x} \in \mathcal{X}$, the covariance of an approximate posterior sample $\hat{f}_t(\boldsymbol{x})$ is given by*

$$Cov\big[\hat{f}_t(\boldsymbol{x})\big] = \hat{\sigma}_t(\boldsymbol{x})^2 \tag{21}$$

*where*

$$\begin{aligned}
\hat{\sigma}_t(\boldsymbol{x})^2 = &\frac{\eta_\beta^2}{M}\varphi(\boldsymbol{W}\boldsymbol{x})\varphi(\boldsymbol{W}\boldsymbol{x})^\top \\
&- \frac{\eta_\beta^2}{M}\varphi(\boldsymbol{W}\boldsymbol{x})\varphi(\boldsymbol{X}\boldsymbol{W}^\top)^\top\big(k(\boldsymbol{X},\boldsymbol{X})+\epsilon^2\boldsymbol{I}_{N\times N}\big)^{-1}k(\boldsymbol{X},\boldsymbol{x}) \\
&- k(\boldsymbol{x},\boldsymbol{X})\big(k(\boldsymbol{X},\boldsymbol{X})+\epsilon^2\boldsymbol{I}_{N\times N}\big)^{-1}\frac{\eta_\beta^2}{M}\varphi(\boldsymbol{X}\boldsymbol{W}^\top)\varphi(\boldsymbol{W}\boldsymbol{x})^\top \\
&+ k(\boldsymbol{x},\boldsymbol{X})\big(k(\boldsymbol{X},\boldsymbol{X})+\epsilon^2\boldsymbol{I}_{N\times N}\big)^{-1}\Big(\frac{\eta_\beta^2}{M}\varphi(\boldsymbol{X}\boldsymbol{W}^\top)\varphi(\boldsymbol{W}\boldsymbol{X}^\top)+\epsilon^2\boldsymbol{I}_{N\times N}\Big)\dots \\
&\quad \big(k(\boldsymbol{X},\boldsymbol{X})+\epsilon^2\boldsymbol{I}_{N\times N}\big)^{-1}k(\boldsymbol{X},\boldsymbol{x}).
\end{aligned}$$

*Proof.* Although this lemma is not explicitly used later, we wish to show the reader that the true covariance is recovered as number of random features $M$ grows large. For input vector $\boldsymbol{x} \in \mathbb{R}^d$, and $N$ data points stored in $\boldsymbol{X} \in \mathbb{R}^{N\times d}$ and $\boldsymbol{y} \in \mathbb{R}^N$, with noise $\boldsymbol{\epsilon} \sim \mathcal{N}(\boldsymbol{0}_N, \epsilon^2\boldsymbol{I}_{N\times N})$ and $k(\boldsymbol{x}, \boldsymbol{X}) \in \mathbb{R}^N$, our posterior sample is expressed as

$$\hat{f}_t(\boldsymbol{x}) = \underbrace{\hat{f}(\boldsymbol{x})}_{\text{prior}} + \underbrace{k(\boldsymbol{x},\boldsymbol{X})^\top\big(k(\boldsymbol{X},\boldsymbol{X})+\epsilon^2\boldsymbol{I}_{N\times N}\big)^{-1}\big(\boldsymbol{y}-\hat{f}(\boldsymbol{X})-\boldsymbol{\epsilon}\big)}_{\text{posterior update}},$$

where the prior term is a sum of $M$ random features expressed as

$$\hat{f}(\boldsymbol{x}) = \boldsymbol{\beta}^\top\varphi(\boldsymbol{W}\boldsymbol{x})$$

for some fixed $\boldsymbol{W} \in \mathbb{R}^{M\times d}$, $\boldsymbol{\beta} \sim \mathcal{N}(\boldsymbol{0}_M, \frac{\eta_\beta^2}{M})$ and feature mapping $\varphi$ (applied element-wise). The mean is given by

$$\mu_{t-1}(\boldsymbol{x}) = k(\boldsymbol{x},\boldsymbol{X})\big(k(\boldsymbol{X},\boldsymbol{X})+\epsilon^2\boldsymbol{I}_{N\times N}\big)^{-1}\boldsymbol{y}.$$

Then

$$\begin{aligned}
\text{Cov}\big[\hat{f}_t(\boldsymbol{x})\big] &= \mathbb{E}\Big[\big(\hat{f}_t(\boldsymbol{x})-\mu_{t-1}(\boldsymbol{x})\big)\big(\hat{f}_t(\boldsymbol{x})-\mu_{t-1}(\boldsymbol{x})\big)^\top\Big] \\
&= \mathbb{E}\Big[\big(\hat{f}(\boldsymbol{x})-k(\boldsymbol{x},\boldsymbol{X})\big(k(\boldsymbol{X},\boldsymbol{X})+\epsilon^2\boldsymbol{I}_{N\times N}\big)^{-1}\big(\hat{f}(\boldsymbol{X})+\boldsymbol{\epsilon}\big)\big)\dots \\
&\qquad \big(\hat{f}(\boldsymbol{x})-k(\boldsymbol{x},\boldsymbol{X})\big(k(\boldsymbol{X},\boldsymbol{X})+\epsilon^2\boldsymbol{I}_{N\times N}\big)^{-1}\big(\hat{f}(\boldsymbol{X})+\boldsymbol{\epsilon}\big)\big)^\top\Big] \\
&= \mathbb{E}\Big[\big(\boldsymbol{\beta}^\top\varphi(\boldsymbol{W}x)-k(\boldsymbol{x},\boldsymbol{X})\big(k(\boldsymbol{X},\boldsymbol{X})+\epsilon^2\boldsymbol{I}_{N\times N}\big)^{-1}\big(\varphi(\boldsymbol{X}\boldsymbol{W}^\top)\boldsymbol{\beta}+\boldsymbol{\epsilon}\big)\big)\dots \\
&\qquad \big(\boldsymbol{\beta}^\top\varphi(\boldsymbol{W}\boldsymbol{x})-k(\boldsymbol{x},\boldsymbol{X})\big(k(\boldsymbol{X},\boldsymbol{X})+\epsilon^2\boldsymbol{I}_{N\times N}\big)^{-1}\big(\varphi(\boldsymbol{X}\boldsymbol{W}^\top)\boldsymbol{\beta}+\boldsymbol{\epsilon}\big)\big)^\top\Big] \\
&= \frac{\eta_\beta^2}{M}\varphi(\boldsymbol{W}\boldsymbol{x})\varphi(\boldsymbol{W}\boldsymbol{x})^\top \\
&\quad - \frac{\eta_\beta^2}{M}\varphi(\boldsymbol{W}\boldsymbol{x})\varphi(\boldsymbol{X}\boldsymbol{W}^\top)^\top\big(k(\boldsymbol{X},\boldsymbol{X})+\epsilon^2\boldsymbol{I}_{N\times N}\big)^{-1}k(\boldsymbol{X},\boldsymbol{x}) \\
&\quad - k(\boldsymbol{x},\boldsymbol{X})\big(k(\boldsymbol{X},\boldsymbol{X})+\epsilon^2\boldsymbol{I}_{N\times N}\big)^{-1}\frac{\eta_\beta^2}{M}\varphi(\boldsymbol{X}\boldsymbol{W}^\top)\varphi(\boldsymbol{W}\boldsymbol{x})^\top \\
&\quad + k(\boldsymbol{x},\boldsymbol{X})\big(k(\boldsymbol{X},\boldsymbol{X})+\epsilon^2\boldsymbol{I}_{N\times N}\big)^{-1}\Big(\frac{1}{M}\varphi(\boldsymbol{X}\boldsymbol{W}^\top)\varphi(\boldsymbol{W}\boldsymbol{X}^\top)+\epsilon^2\boldsymbol{I}_{N\times N}\Big)\dots \\
&\qquad \big(k(\boldsymbol{X},\boldsymbol{X})+\epsilon^2\boldsymbol{I}_{N\times N}\big)^{-1}k(\boldsymbol{X},\boldsymbol{x})
\end{aligned}$$

as required. Although this expression is unwieldy, note that as $M \to \infty$, $\frac{\eta_\beta^2}{M}\varphi(\boldsymbol{W}\boldsymbol{x})\varphi(\boldsymbol{W}\boldsymbol{x}) \to k(\boldsymbol{x}, \boldsymbol{x})$ and $\frac{\eta_\beta^2}{M}\varphi(\boldsymbol{X}\boldsymbol{W}^\top)\varphi(\boldsymbol{W}\boldsymbol{X}^\top) \to k(\boldsymbol{X}, \boldsymbol{X})$, allowing the final two terms to cancel, recovering the true covariance. $\quad\square$

**Lemma B.4.** *Given observation data $\mathcal{D}_{t-1}$, and that $E^f(t)$ is true (lemma Theorem B.1), then for every $\boldsymbol{x} \in \mathcal{X}$,*

$$\Pr\left(f_t(\boldsymbol{x}) > f(\boldsymbol{x})\right) > \frac{1}{4e\sqrt{\pi}}. \tag{22}$$

*Proof.* We make use of the Gaussian anti-concentration lemma, stating that for $X \sim \mathcal{N}(\mu, \sigma^2)$, and $\beta > 0$,

$$\Pr\left(\frac{X - \mu}{\sigma} > \beta\right) \geq \frac{e^{-\beta^2}}{4\sqrt{\pi}\beta}.$$

Given $f_t(\boldsymbol{x}) \sim \mathcal{N}(\mu_{t-1}(\boldsymbol{x}), \beta_t^2\sigma_{t-1}(\boldsymbol{x})^2)$, and taking $\beta = 1$, we have that

$$\Pr\left(f_t(\boldsymbol{x}) - \mu_{t-1}(\boldsymbol{x}) > \beta_t\sigma_{t-1}(\boldsymbol{x})\right) \geq \frac{1}{4e\sqrt{\pi}}$$

Conditioning on lemma Theorem B.1's bound, we have that

$$\begin{aligned} f_t(\boldsymbol{x}) - \mu_{t-1}(\boldsymbol{x}) &> \beta_t\sigma_{t-1}(\boldsymbol{x}) \\ &\geq |f(\boldsymbol{x}) - \mu_{t-1}(\boldsymbol{x})| \end{aligned}$$

implying

$$f_t(\boldsymbol{x}) > f(\boldsymbol{x})$$

with probability greater than $1/(4e\sqrt{\pi})$, as required. $\quad\square$

**Definition B.5.** Let $c_t = \beta_t(1 + \sqrt{2\log(|\mathcal{X}_t|t^2)})$. Then define the set of *saturated* points in discretisation $\mathcal{X}_t$ at iteration $t$ as

$$S_t = \{\boldsymbol{x} \in \mathcal{X}_t : \Delta(\boldsymbol{x}) > c_t\sigma_{t-1}(\boldsymbol{x})\} \tag{23}$$

where $\Delta(\boldsymbol{x}) = f(\boldsymbol{x}^*) - f(\boldsymbol{x})$ is instantaneous regret and $\boldsymbol{x}^* = \arg\max_{\boldsymbol{x} \in \mathcal{X}'} f(\boldsymbol{x})$. This collects points $\boldsymbol{x}$ such that their instantaneous regret is sufficiently large. Conversely, we denote the set of *unsaturated points* by

$$S_t^{\complement} = \{\boldsymbol{x} \in \mathcal{X}_t \setminus S_t\}. \tag{24}$$

**Lemma B.6.** *Let $\boldsymbol{x}_t = \arg\max_{\boldsymbol{x} \in \mathcal{X}_t} f_t(\boldsymbol{x})$ be the query point selected during round $t$. Then for any dataset $\mathcal{D}_{t-1}$, conditioned on $E^f(t)$ (lemma Theorem B.1),*

$$\Pr\left(\boldsymbol{x}_t \in S_t^{\complement} \mid \mathcal{D}_{t-1}\right) > \frac{1}{4e\sqrt{\pi}} - \frac{1}{t^2}. \tag{25}$$

*Proof.* Here, as is standard in Bayesian optimisation regret proofs, we assume that $\arg\max_{\boldsymbol{x} \in \mathcal{X}_t} f_t(\boldsymbol{x})$ is able to be found exactly. This is often difficult to guarantee, due to the nonconvexity of $f_t$. In reality, it may only be possible to guarantee a local stationary point of $f_t$. We discuss this tension in the main paper. For now, observe that if $f_t(\boldsymbol{x}^*) > f_t(\boldsymbol{x})$ for all saturated points $\boldsymbol{x} \in S_t$ (noting that $\boldsymbol{x}^*$ is always unsaturated), then $\boldsymbol{x}_t$ will be unsaturated. This is because at least one unsaturated input ($\boldsymbol{x}^*$) has a larger value of $f_t$ than all saturated inputs, and so $f_t(\boldsymbol{x}_t)$ will be at least this. Therefore

$$\Pr\left(\boldsymbol{x}_t \in S_t^{\complement} \mid \mathcal{D}_{t-1}\right) \geq \Pr\left(f_t(\boldsymbol{x}^*) > f_t(\boldsymbol{x}), \forall \boldsymbol{x} \in S_t \mid \mathcal{D}_{t-1}\right). \tag{26}$$

Conditioning on both $E^f(t)$ (lemma Theorem B.1) and $E^{f_t}(t)$ (lemma Theorem B.2) we have that

$$
\begin{aligned}
|f(\boldsymbol{x}) - f_t(\boldsymbol{x})| &\leq |f(\boldsymbol{x}) - \mu_{t-1}(\boldsymbol{x})| + |\mu_{t-1}(\boldsymbol{x}) - f_t(\boldsymbol{x})| \\
&= \beta_t \sigma_{t-1}(\boldsymbol{x}) + \beta_t \sqrt{2\log|\mathcal{X}_t|t^2}\sigma_{t-1}(\boldsymbol{x}) \\
&= c_t \sigma_{t-1}(\boldsymbol{x})
\end{aligned}
\tag{27}
$$

where $c_t = \beta_t(1 + \sqrt{2\log|\mathcal{X}_t|t^2})$. Recalling the definition of saturated points (Theorem B.5), it then follows that

$$
\begin{aligned}
f_t(\boldsymbol{x}) &\leq f(\boldsymbol{x}) + c_t \sigma_{t-1}(\boldsymbol{x}) \\
&\leq f(\boldsymbol{x}) + \Delta(\boldsymbol{x}) \\
&= f(\boldsymbol{x}) + f(\boldsymbol{x}^*) - f(\boldsymbol{x}) \\
&= f(\boldsymbol{x}^*),
\end{aligned}
$$

which implies

$$
\Pr\left(f_t(\boldsymbol{x}^*) > f_t(\boldsymbol{x}), \forall \boldsymbol{x} \in S_t \,|\, \mathcal{D}_{t-1}, E^{f_t}(t)\right) \geq \Pr\left(f_t(\boldsymbol{x}^*) > f(\boldsymbol{x}^*) \,|\, \mathcal{D}_{t-1}, E^{f_t}(t)\right).
\tag{28}
$$

Combining (26) with (28) and following Dai et al. (2020), we have that

$$
\begin{aligned}
\Pr\left(\boldsymbol{x}_t \in S_t^{\complement} \,|\, \mathcal{D}_{t-1}\right) &\geq \Pr\left(f_t(\boldsymbol{x}^*) > f_t(\boldsymbol{x}) \,|\, \mathcal{D}_{t-1}, E^{f_t}(t)\right) \\
&\geq \Pr\left(f_t(\boldsymbol{x}^*) > f(\boldsymbol{x}^*) \,|\, \mathcal{D}_{t-1}\right) - \Pr\left(\overline{E^{f_t}(t)} \,|\, \mathcal{D}_{t-1}\right) \\
&\geq \frac{1}{4e\sqrt{\pi}} - \frac{1}{t^2},
\end{aligned}
$$

which follows from Theorem B.4 and that $\overline{E^{f_t}(t)}$ occurs with probability less than $1/t^2$ (converse of Theorem B.2). $\qquad\square$

**Lemma B.7.** *Let $r_t = f(\boldsymbol{x}^*) - f(\boldsymbol{x}_t)$ be the instantaneous regret (simple regret). Then for data $\mathcal{D}_{t-1}$, working in discretisation $\mathcal{X}_t$, conditioned on the event $E^{f_t}(t)$,*

$$
\mathbb{E}\left[r_t \,|\, \mathcal{D}_{t-1}\right] \leq c_t\left(1 + 40e\sqrt{\pi}\right)\mathbb{E}\left[\sigma_{t-1}(\boldsymbol{x}_t) \,|\, \mathcal{D}_{t-1}\right] + \frac{2B}{t^2}.
\tag{29}
$$

*Proof.* Let $\overline{\boldsymbol{x}}_t$ refer to the unsaturated input at time $t$ with the smallest standard deviation, that is,

$$
\overline{\boldsymbol{x}}_t = \arg\min_{\boldsymbol{x} \in S_t^{\complement}} \sigma_{t-1}(\boldsymbol{x}).
\tag{30}
$$

Then, conditioned on $E^{f_t}(t)$, for any data $\mathcal{D}_{t-1}$, by the law of total expectation we have

$$
\begin{aligned}
\mathbb{E}\left[\sigma_{t-1}(\boldsymbol{x}_t) \,|\, \mathcal{D}_{t-1}\right] &\geq \mathbb{E}\left[\sigma_{t-1}(\boldsymbol{x}_t) \,|\, \mathcal{D}_{t-1}, \boldsymbol{x}_t \in S_t^{\complement}\right]\Pr\left(\boldsymbol{x}_t \in S_t^{\complement} \,|\, \mathcal{D}_{t-1}\right) \\
&\geq \sigma_{t-1}(\overline{\boldsymbol{x}}_t)\left(\frac{1}{4e\sqrt{\pi}} - \frac{1}{t^2}\right) \\
&= \sigma_{t-1}(\overline{\boldsymbol{x}}_t)P_t
\end{aligned}
\tag{31}
$$

where we have used the definition of $\overline{\boldsymbol{x}}_t$ and lemma Theorem B.6, and have denoted the final parentheses term by $P_t$. Next, conditioning on $E^f(t)$ and $E^{f_t}(t)$, we write the instantaneous regret as

$$
\begin{aligned}
r_t = \Delta(\boldsymbol{x}_t) &= f(\boldsymbol{x}^*) - f(\overline{\boldsymbol{x}}_t) + f(\overline{\boldsymbol{x}}_t) - f(\boldsymbol{x}_t) \\
&\leq \underbrace{\Delta(\overline{\boldsymbol{x}}_t)}_{\text{regret def}^n} + \underbrace{f_t(\overline{\boldsymbol{x}}_t) + c_t\sigma_{t-1}(\overline{\boldsymbol{x}}_t) - f_t(\boldsymbol{x}_t) + c_t\sigma_{t-1}(\boldsymbol{x}_t)}_{\text{via (27)}} \\
&\leq \underbrace{c_t\sigma_{t-1}(\overline{\boldsymbol{x}}_t)}_{\text{via } \overline{\boldsymbol{x}}_t \text{ unsaturated}} + c_t\sigma_{t-1}(\overline{\boldsymbol{x}}_t) + c_t\sigma_{t-1}(\boldsymbol{x}_t) + f_t(\overline{\boldsymbol{x}}_t) - f_t(\boldsymbol{x}_t) \\
&= c_t(2\sigma_{t-1}(\overline{\boldsymbol{x}}_t) + \sigma_{t-1}(\boldsymbol{x}_t)) + f_t(\overline{\boldsymbol{x}}_t) - f_t(\boldsymbol{x}_t) \\
&\leq c_t(2\sigma_{t-1}(\overline{\boldsymbol{x}}_t) + \sigma_{t-1}(\boldsymbol{x}_t)),
\end{aligned}
$$

where in the last inequality we observe that the definition of $\boldsymbol{x}_t = \arg\max_{\boldsymbol{x}\in\mathcal{X}'} f_t(\boldsymbol{x})$ implies $f_t(\overline{\boldsymbol{x}}_t) - f_t(\boldsymbol{x}_t) \leq 0$. Next, given that instantaneous regret is $2B$ in the worst case (as $|f(\boldsymbol{x})| < B$), we again utilise law of total expectation to write

$$\mathbb{E}\big[r_t \,|\, \mathcal{D}_{t-1}\big] \leq \mathbb{E}\big[c_t(2\sigma_{t-1}(\overline{\boldsymbol{x}}_t) + \sigma_{t-1}(\boldsymbol{x}_t)) \,|\, \mathcal{D}_{t-1}, E^{f_t}(t)\big] + 2B\Pr\big(\overline{E^{f_t}(t)} \,|\, \mathcal{D}_{t-1}\big)$$

$$\leq \mathbb{E}\big[c_t\big(2\sigma_{t-1}(\overline{\boldsymbol{x}}_t) + \sigma_{t-1}(\boldsymbol{x}_t)\big) \,|\, \mathcal{D}_{t-1}, E^{f_t}(t)\big] + \underbrace{\frac{2B}{t^2}}_{\text{via Theorem B.2}}$$

$$= 2c_t\mathbb{E}\big[\sigma_{t-1}(\overline{\boldsymbol{x}}_t) \,|\, \mathcal{D}_{t-1}, E^{f_t}(t)\big] + c_t\mathbb{E}\big[\sigma_{t-1}(\boldsymbol{x}_t) \,|\, \mathcal{D}_{t-1}, E^{f_t}(t)\big] + \frac{2B}{t^2}$$

$$\leq 2c_t\underbrace{\frac{1}{P_t}\mathbb{E}\big[\sigma_{t-1}(\boldsymbol{x}_t) \,|\, \mathcal{D}_{t-1}, E^{f_t}(t)\big]}_{\text{via (31)}} + c_t\mathbb{E}\big[\sigma_{t-1}(\boldsymbol{x}_t) \,|\, \mathcal{D}_{t-1}, E^{f_t}(t)\big] + \frac{2B}{t^2}$$

$$= c_t\left(1 + \frac{2}{P_t}\right)\mathbb{E}\big[\sigma_{t-1}(\boldsymbol{x}_t) \,|\, \mathcal{D}_{t-1}, E^{f_t}(t)\big] + \frac{2B}{t^2}. \tag{32}$$

Denoting $p = (4e\sqrt{\pi})^{-1}$, we note that for $t \geq 1$,

$$\frac{2}{P_t} = \frac{2}{p - \frac{1}{t^2}} \leq \frac{10}{p},$$

allowing us to write (32) as

$$\mathbb{E}\big[r_t \,|\, \mathcal{D}_{t-1}\big] \leq c_t\left(\frac{10}{p}\right)\mathbb{E}\big[\sigma_{t-1}(\boldsymbol{x}_t) \,|\, \mathcal{D}_{t-1}, E^{f_t}(t)\big] + \frac{2B}{t^2},$$

completing the proof. $\qquad\square$

**Definition B.8.** Let $Y_0 = 0$. Let $\mathbb{I}\{\cdot\}$ denote the indicator function. For all $t = 1, \ldots, T$, define

$$\overline{r}_t = r_t\mathbb{I}\{E^{f_t}(t)\},$$

$$X_t = \overline{r}_t - c_t\big(1 + 40e\sqrt{\pi}\big)\sigma_{t-1}(\boldsymbol{x}_t) - \frac{2B}{t^2},$$

$$Y_t = \sum_{s=1}^{t} X_s.$$

**Lemma B.9.** *Conditioned on lemma Theorem B.7 (which would occur with probability $\geq 1 - \delta/2$), $Y_t : t = 0, \ldots, T$ is a super-martingale with respect to $\mathcal{D}_t$.*

*Proof.* By definition,

$$\mathbb{E}\big[Y_t - Y_{t-1} \,|\, \mathcal{D}_{t-1}\big] = \mathbb{E}\big[X_t \,|\, \mathcal{D}_{t-1}\big]$$

$$= \mathbb{E}\left[\overline{r}_t - c_t\big(1 + 40e\sqrt{\pi}\big)\sigma_{t-1}(\boldsymbol{x}_t) - \frac{2B}{t^2} \,|\, \mathcal{D}_{t-1}\right]$$

$$= \mathbb{E}\big[\overline{r}_t \,|\, \mathcal{D}_{t-1}\big] - \left[c_t\big(1 + 40e\sqrt{\pi}\big)\mathbb{E}\big[\sigma_{t-1}(\boldsymbol{x}_t) \,|\, \mathcal{D}_{t-1}\big] + \frac{2B}{t^2}\right]$$

$$\leq 0.$$

The inequality holds due to lemma Theorem B.7 when $E^{f_t}(t)$ is true. When $E^{f_t}(t)$ is false, it holds trivially, as $\overline{r}_t = 0$ by definition. $\qquad\square$

**Lemma B.10.** (Azuma-Hoeffding inequality). *For any $\delta' \in (0, 1)$, if a super-martingale $Z_T : t = 1, \ldots, T$ satisfies $|Z_t - Z_{t-1}| \leq \alpha_t$ for some constant $\alpha_t$, then for all $t = 1, \ldots, T$,*

$$Z_T - Z_0 \leq \sqrt{2\log(1/\delta')\sum_{t=1}^{T}\alpha_t^2}$$

*holds with probability $\geq 1 - \delta'$.*

**Lemma B.11.** *For $\delta \in (0, 1)$,*

$$R_T \leq c_T(1 + 40e\sqrt{\pi})\mathcal{O}\big(\sqrt{T\gamma_T}\big) + \frac{B\pi^2}{3} + \big[c_T(1 + 16Be\sqrt{\pi} + 40e\sqrt{\pi}) + \mathcal{O}\big(\sqrt{\log T}\big)\big]\sqrt{2T\log\frac{4}{\delta}}$$

*holds with probability $\geq 1 - \delta$, where $R_T = \sum_{t=1}^{T} r_t$ is cumulative regret and $\gamma_T$ is the maximum information gain about $f$ obtained from any set of $T$ observations.*

*Proof.* Note that by definition, $|\bar{r}_t| < 2B$. Given a compact domain, we upper bound $\sigma_{t-1}(\boldsymbol{x}_t)$ by 1 without loss of generality. Recall $c_t = \beta_t(1 + \sqrt{2\log|\mathcal{X}_t|t^2})$, and observe that $c_t \geq 4e\sqrt{\pi} \geq 4e\sqrt{\pi}/t^2$. Then

$$|Y_t - Y_{t-1}| = |X_t| = \Big|\bar{r}_t - c_t(1 + 40e\sqrt{\pi})\sigma_{t-1}(\boldsymbol{x}_t) - \frac{2B}{t^2}\Big| \tag{33}$$

$$\leq |\bar{r}_t| + c_t(1 + 40e\sqrt{\pi})\sigma_{t-1}(\boldsymbol{x}_t) + \frac{2B}{t^2}$$

$$\leq 2B + c_t(1 + 40e\sqrt{\pi}) + \frac{2B}{t^2}$$

$$\leq 2Bc_t4e\sqrt{\pi} + c_t(1 + 40e\sqrt{\pi}) + 2Bc_t4e\sqrt{\pi}$$

$$= c_t\big(1 + (4B + 10)4e\sqrt{\pi}\big).$$

This implies, by the Azuma-Hoeffding inequality (lemma Theorem B.10, taking $\delta' = \delta/4$), and recalling $Y_0 = 0$, that

$$Y_T \leq \sqrt{2\log(4/\delta)\sum_{t=1}^{T}\Big[c_t\big(1 + (4B + 10)4e\sqrt{\pi}\big)\Big]^2}$$

holds with probability $\geq 1 - \delta/4$. By definition, and summing over $t$, we have

$$\sum_{t-1}^{T}\bar{r}_t = \sum_{t=1}^{T}c_t\big(1 + 40e\sqrt{\pi}\big)\sigma_{t-1}(\boldsymbol{x}_t) + \sum_{t=1}^{T}\frac{2B}{t^2} + \sum_{t=1}^{T}\big[Y_t - Y_{t-1}\big]$$

$$= \sum_{t=1}^{T}c_t\big(1 + 40e\sqrt{\pi}\big)\sigma_{t-1}(\boldsymbol{x}_t) + \sum_{t=1}^{T}\frac{2B}{t^2} + Y_T$$

$$\leq \sum_{t=1}^{T}c_t\big(1 + 40e\sqrt{\pi}\big)\sigma_{t-1}(\boldsymbol{x}_t) + \sum_{t=1}^{T}\frac{2B}{t^2} + \sqrt{2\log(4/\delta)\sum_{t=1}^{T}\Big[c_t\big(1 + (4B + 10)4e\sqrt{\pi}\big)\Big]^2}.$$

Next, noting that $c_t$ is increasing in $t$, and that $\sum_{t=1}^{T} 1/t^2 \leq \pi^2/6$. Note that $\sum_{t=1}^{T}\sigma_{t-1}(\boldsymbol{x}_t) = \mathcal{O}\big(\sqrt{T\gamma_T}\big)$, shown in Srinivas et al. (2010, lemmas 5.3 and 5.4). It follows that

$$\sum_{t-1}^{T}\bar{r}_t \leq c_T\big(1 + 40e\sqrt{\pi}\big)\mathcal{O}\big(\sqrt{T\gamma_T}\big) + \frac{B\pi^2}{3} +$$

$$\Big[c_T\big(1 + (4B + 10)4e\sqrt{\pi}\big)\Big]\sqrt{2T\log(4/\delta)}$$

holds with probability greater than $1 - \delta/4$. $\qquad\square$

**Regret bound.** Firstly, note that (by definition),

$$c_t = \beta_t(1 + \sqrt{2\log(|\mathcal{X}_t|t^2)})$$

$$= \big(B + \epsilon\sqrt{2(\gamma_{t-1} + 1 + \log(4/\delta))}\big)\big(1 + \sqrt{2\log(|\mathcal{X}_t|t^2)}\big)$$

$$= \mathcal{O}\Big(\big(B + \sqrt{\gamma_t + \log(1/\delta)}\big)\sqrt{\log t}\Big).$$

Then, by lemma Theorem B.11, it follows that

$$R_T = \mathcal{O}\Big( \big(B + \sqrt{\gamma_T + \log(1/\delta)}\big)\sqrt{\log T}\sqrt{T\gamma_T} +$$

$$\big(B+1\big)\big(B + \sqrt{\log(1/\delta)}\big)\sqrt{\log T}\sqrt{T\log(1/\delta)}\Big)$$

$$= \mathcal{O}\Big( \big(B+1\big)\sqrt{T\gamma_T \log T \log(1/\delta)\big(\gamma_T + \log(1/\delta)\big)}\Big)$$

$$= \tilde{\mathcal{O}}\Big( \big(B+1\big)\gamma_T\sqrt{T}\Big).$$

## C   Universal DC decompositions of smooth functions

Difference-of-convex decompositions (5) are not unique. For any DC function $g(\boldsymbol{x}) = g_1(\boldsymbol{x}) - g_2(\boldsymbol{x})$ and some convex function $c(\boldsymbol{x})$, observe that

$$g(\boldsymbol{x}) = \underbrace{g_1(\boldsymbol{x}) + c(\boldsymbol{x})}_{\text{convex}} - \underbrace{\big(g_2(\boldsymbol{x}) + c(\boldsymbol{x})\big)}_{\text{convex}}$$

also forms a DC decomposition of $g$. Under this formulation, even if $g$ is not naturally DC, we may be able to choose an appropriate $c(\boldsymbol{x})$ such that $g(\boldsymbol{x}) + c(\boldsymbol{x})$ is convex, giving us a (perhaps "artificial") DC decomposition of $g$ of the form

$$g(\boldsymbol{x}) = \underbrace{g(\boldsymbol{x}) + c(\boldsymbol{x})}_{\text{convex}} - \underbrace{c(\boldsymbol{x})}_{\text{convex}}.$$

For a GP sample constructed from random features with bounded second derivative, this may be accomplished by simply taking $c$ to be a quadratic with an appropriate coefficient. We apply this to a GP sample constructed with random cosine features, which corresponds to a GP with squared exponential kernel. This GP sample will take the form

$$f(\boldsymbol{x}) = \boldsymbol{\beta}^\top \cos(\boldsymbol{W}\boldsymbol{x} + \boldsymbol{b}),$$

where cos is applied element-wise, as per Rahimi & Recht (2007). Rather than using pathwise updating, we sample from the posterior by doing Bayesian regression on $\boldsymbol{\beta}$. Given the cosine function has a bounded second derivative, we may obtain a DC decomposition of f by writing

$$f(\boldsymbol{x}) = \underbrace{\boldsymbol{\beta}^\top \cos(\boldsymbol{W}\boldsymbol{x} + \boldsymbol{b}) + c\|\boldsymbol{x}\|^2}_{\text{convex}} - \underbrace{c\|\boldsymbol{x}\|^2}_{\text{convex}},$$

where

$$c > \frac{1}{2}\sum_{i=1}^{M} |\beta_i| \|\boldsymbol{w}_i\boldsymbol{w}_i^\top|,$$

and $\boldsymbol{w}_i$ indicates the $i$th row of $\boldsymbol{W}$. By constructing the sample like this, we may maximise it using the DC algorithm. However, experimental results show poor performance compared to a) optimising a cosine-features-based sample simply using LBFGS and b) using the DC algorithm on a sample that "naturally" admits a DC decomposition, such as one constructed from random ReLU features. More generally, any $L$-smooth function may be expressed as DC in this way when we take $c > L/2$.

## D   Notation

Vectors are written in boldface lowercase, and matrices in boldface uppercase. We denote the objective function by $f$ and use $\boldsymbol{x} = [x_1, \ldots, x_d]^\top$ to denote a $d$-dimensional input vector, with corresponding objective function observation $y$. We stack $N$ input vectors into the rows of a matrix $\boldsymbol{X}$ such that

$\boldsymbol{X} = [\boldsymbol{x}_1^\top; \ldots; \boldsymbol{x}_N^\top] \in \mathbb{R}^{N \times d}$. Similarly, we may stack their corresponding observations into a vector $\boldsymbol{y} = [y_1, \ldots, y_N]^\top$. Let $k : \mathcal{X} \times \mathcal{X} \to \mathbb{R}$ denote a positive semi-definite kernel. We define the two vectors $\boldsymbol{k}(\boldsymbol{x}, \boldsymbol{X}) = [k(\boldsymbol{x}, \boldsymbol{x}_1), \ldots, k(\boldsymbol{x}, \boldsymbol{x}_N)] \in \mathbb{R}^{1 \times N}$ and $\boldsymbol{k}(\boldsymbol{X}, \boldsymbol{x}) = \boldsymbol{k}(\boldsymbol{x}, \boldsymbol{X})^\top \in \mathbb{R}^N$. For $N$ input vectors stacked into $\boldsymbol{X}$, we denote their covariance matrix by $\boldsymbol{K}_N \in \mathbb{R}^{N \times N}$, with the $(i, j)$-th element given by $k(\boldsymbol{x}_i, \boldsymbol{x}_j)$. The identity matrix of size $N \times N$ is denoted by $\boldsymbol{I}_N$. We use $\boldsymbol{\psi}^{(M)}$ to denote a vector-value feature mapping with $M$ outputs. We use $\sigma$ to denote an arbitrary activation function. We use sans-serif font to indicate a random variable (r.v.), and serif font to indicate a deterministic variable. For example, $\mathsf{f}$ is an r.v., while $f$ is deterministic, $\boldsymbol{\beta}$ is an r.v., whereas $\beta$ is deterministic, and $\boldsymbol{\epsilon}$ is an r.v. with $\epsilon$ being deterministic.

