# OpenReview forum: "Bayesian Optimisation via Difference-of-Convex Thompson Sampling"
_TMLR — Rejected by TMLR_

### Review · Reviewer_MWBP · 2025-12-17

**Summary Of Contributions:**

This paper proposes to address the Bayesian problem by integrating difference-of-convex optimization into Thompson sampling. Specifically, it constructs Gaussian Process (GP) priors using random neural network features and pathway updates, which allows the use of the difference-of-convex algorithm (DCA) to solve the acquisition problem. Both the theoretical analysis and empirical studies are provided to show the effectiveness of the proposed method.

The method is novel, but I have the following concerns (I don't work on Bayesian optimization, so my opinions may not be reasonable):
1. Theorem 3.1 assumes that the acquisition problem is solved exactly. However, the DCA only guarantees convergence to a local stationary point.
2. In the proof of Lemma B.4, it seems that the author bounds the scalar $\| \hat{f}\_t(x) - \mu\_{t-1}(x) \|$ using the norm of weight vector $\\| z\\|_2^2$. This introduces a factor of $\sqrt{M}$. Since $M$ must be large, will the regret bound become problematic?
3. DCA introduces an outer loop subproblem. It requires more computation overhead. It would be better to compare the efficiency of the proposed method with baselines.
4. Algorithm 1 suggests that weights $W$ and $\beta$ are redrawn at each iiteration. The implications of this variance on the optimization stability are not discussed.

Therefore, I recommend a major revision.

**Audience:**

Yes

**Audience Explanation:**

The idea of using difference-of-convex optimization in Thompson sampling is novel, and it provides insights to the community.

**Broader Impact Concerns:**

I think there are no ethical concerns.

**Claims And Evidence:**

No

**Claims Explanation:**

As mentioned above, there exists a theoretical gap in Theorem 3.1. Also, clarifications regarding proof in Lemma B.4 should be provided.

**Requested Changes:**

I think more details can be provided.

1. Report the computation cost.
2. Compare the methods with matched compute budgets.

---

> ### Author Response · Authors · 2026-03-19
>
> Thank you for the valuable feedback. We’ve made changes based on reviewer feedback, and these are highlighted in blue in the paper. To answer your questions directly:
>
> 1. Yes, this is the unfortunate nature of standard BO regret proofs - it is required that the subproblem is solved globally and exactly at each round in order to guarantee convergence. Practically, this is difficult to guarantee. In reality, users will optimise the acquisition to the best of their ability, but will most likely not be able to guarantee they are within some tolerance of the global max, and may only be able to find a local stationary point of the acquisition function. Our intention with this paper was to provide a method to make that acquisition optimisation easier by exploiting DC structure. While exploring how inexact acquisition solutions affect overall regret bounds in BO is an important research question, it is beyond the scope of this work. We’ve provided some more clarity around this in the paper itself.
>
> 2. Thank you, this was well spotted and is indeed counter intuitive. We realised a mistake that this holds for the dimension of the vector, NOT the random features M. Regardless, lemmas B.3, B.4 and B.5 were redundant for the main result, and have now been removed.
>
> 3. DCA does introduce an additional convex subproblem, and has more computational overhead compared to LBFGS. DCA can be likened to a vanilla Newton method: where a Newton method will approximate the objective with a quadratic around the incumbent point per iteration, DCA will approximate the objective using the convex component of the objective function itself per iteration. This convex subproblem can then be solved using any local optimiser. However, in a BO setting, it is assumed that computational expense overwhelmingly comes from evaluating the objective function, and these evaluations can span from minutes to days (in the case of large cluster simulations). So while there is some extra computational overhead with DCA, this is generally considered irrelevant (within reason).
>
> 4. You are correct, W is fixed, and \beta is redrawn at each iteration, per the original random Fourier features paper (Rahimi and Recht, 2007). This is stated in the main body but was not clear in algorithm 1, we have now clarified this in the text.

---

### Review · Reviewer_KvMF · 2026-01-20

**Summary Of Contributions:**

The authors tackle the problem of optimising acquisition functions in Bayesian optimisation. They propose using pathwise updates alongside random features to obtain a Gaussian process posterior that admits a difference-of-convex decomposition. This enables more efficient optimisation of the acquisition function in the Thompson Sampling framework. The method performs really well throughout a variety of experiments. However, the main weakness of this method is that it restricts the choice of kernels; in particular, the kernel must be convex. Moreover, the authors focus on the arc-cosine kernel, which requires a specific choice of prior mean.

**Audience:**

Yes

**Audience Explanation:**

Bayesian Optimisation is an active area of research, and the problem acquisition function optimisation is relevant to the community.

**Claims And Evidence:**

Yes

**Claims Explanation:**

As far as I can tell, the claims made in the submission are accurate.

**Requested Changes:**

- In my opinion, the fact that the method needs a specific prior mean to perform correctly must be discussed sooner in the paper, at least in the experiment Section. In this line, I'm curious to see the impact of the prior mean on the results: ReLU with and without prior mean, and Sq exp with and without the prior mean.
- How is $c$ in the prior mean selected for each experiment? I can see in the appendix that the value of $c$ varies substantially across experiments, so I'm curious about how the authors choose it and what impact it has.
- I think it would be helpful if the authors could expand on how restrictive the method is. While the authors claim that the restriction is that the features must be convex, they also need to derive the kernel analytically so that the update step can be computed. Therefore, the method is more restricted in both the kernel and the prior mean.
- Figure 4 is not discussed in the paper, which, in my opinion, seems important, since it shows that the results in Table 1 are mostly due to the choice of kernel if I am understanding correctly.
Minor comments:
- Author's comment left in appendix B, paragraph "Design optimisation and granular simulations".

---

> ### Author Response · Authors · 2026-03-19
>
> Thank you for the valuable feedback. We’ve made changes based on reviewer feedback, and these are highlighted in blue in the paper.
>
> We’ve added discussion on the prior mean and moved it further up in the paper. The main idea is that because the kernel is non mean-recurring (samples will not necessarily revert to the mean away from data), we can find better performance by damping these large sample values on the edge of the domain, because we usually intuit that the function maximiser will be somewhere in the interior. In our experiments, we selected c heuristically, and have added some sensitivity experiments into the main paper.
>
> We’ve added some more discussion on Figure 4 regarding how restrictive the kernel choice really is. Yes, the method is restricted to convex random features and their resulting limiting kernels. For our case, we use ReLUs with the corresponding arc-cosine kernel. Figure 4 shows an example where this models the objective better than a squared exponential kernel. The intention here is to show that while kernel choice is indeed more limited, this does not necessarily make for a worse model. The limiting kernel does not necessarily need to be derived analytically - it can be approximated using the features themselves via equation (7).
>
> Thank you for spotting the author comment, this has now been removed.

---

### Review · Reviewer_RxMQ · 2026-02-05

**Summary Of Contributions:**

The paper introduces a novel way to find a new candidate with the Thompson sampling acquisition function in Bayesian optimization. The method is based on the so-called difference-of-convex method, that supposedly works better than standard gradient-based optimization. In experiments using 4 synthetic functions and 2 more realistic problems, the new method outperforms previous Thompson sampling approaches and also EI and UCB approaches.

**Additional Comments:**

Open Questions:
* How is the theoretical analysis in Section 3.2 necessary to derive the Algorithms 1 and 2? I understand that the result is interesting by itself, but it is not picked up again in Sections 3.3 and 3.4.
* Is the only difference between ReLU (DC, ours) and ReLU (no DC) the local optimization mechanism after finding an initial point to run a local optimization algorithm? So, the difference is to use the DC optimizer in Algorithm 2 vs LBFGS?
* Figure 4 makes me wonder if the paper is not only about the optimization, but also about the model? If we use ReLU random fourier features, this would lead to a different Gaussian process. Shouldn't this also be compared to see if the improvements actually come from using a different prior?

**Audience:**

Yes

**Audience Explanation:**

Bayesian optimization is the workhorse for global optimization in machine learning, used by many researchers and practitioners. Any improvements in Bayesian optimization would directly lead to downstream improvements.

**Broader Impact Concerns:**

None.

**Claims And Evidence:**

No

**Claims Explanation:**

The claim of improved performance is demonstrated using small experiments. Unfortunately, there are not enough details to understand the baselines. The paper also provides a new theoretical analysis of Thompson sampling, but I am unsure how this connects to the difference-of-convex methodology.

**Requested Changes:**

Major:
* Move the quadratic bowl prior mean to the methodology, and do not hide it under limitations. Also, it would be important perform an ablation without such a prior would be interesting. I thought a lot about whether the paper should also show the performance of EI with a quadratic bowl prior, but because this paper is mostly about improving Thompson sampling, I do not think this is vital. However, if the authors want to be complete, they could go ahead and add such a baseline.
* Conduct a sensitivity analysis of the hyperparameter c of the bowl prior. As I can see, the different experiments rely on a different value of c, which would render the method impractical to use.
* Improve the description of the BO loop. The paper is ambiguous about whether \beta is increased or not. This should be rather clear. Also, the kernel for the Gaussian processes is not mentioned.
* The paper does not include important baselines:
    1. Thompson sampling baseline. The paper should contain a baseline of sampling from a function from the GP at a random set of points (let's says 2000-5000, whatever is feasible in practice) and pick the one with the maximum function value (for a different set of points in every iteration). Thompson sampling is a relevant baseline, as the paper proposes an improved Thompson sampling scheme.
    2. Thompson sampling with reliable, differentiable draws, following Wilson et al. (2020). The paper builds on the Gaussian process model by Wilson et al. (2020), but does not compare against it. As Wilson et al. show, their improved GP sampling scheme can outperform the RFF baseline of this work. Thus, it would be important to know if the algorithm proposed in this paper can actually beat the intermediate method from 2020.
* The DC optimizer in Algorithm 2 appears to be underspecified. There is an argmin in Algorithm 2, but I cannot find how this is optimized.
* The paper only sparingly motivates choices and explains details. I find this rather suboptimal for a journal paper, especially if there are several more pages that could be used. I would love to better understand the difference-of-convex algorithm, and why we should consider it. Therefore, I think some 1d-plots on how the optimization works in practice would be great.

Minor:
* It would be great if the paper could cite an overview/survey of hyperparameter optimization in the introduction in addition to mentioning three individual (important) contributions.
* There is a broken reference on Page 15.
* The reference to Figure 1 in the text should be improved. Currently, it is not part of a sentence.
* Why are lemmas 3-5 needed if they do not influence the result?

---

> ### Author Response · Authors · 2026-03-19
>
> Thank you for your in depth review, we found the feedback very valuable. We’ve made changes based on reviewer feedback, and these are highlighted in blue in the paper.
>
> We have moved the discussion about c further up into the main section, along with some more discussion and references regarding BO hyperparameter selection. Kernels and hyperparameters in BO are often chosen intuitively and heuristically, we’ve provided some more clarity around the motivation for using the quadratic bowl, and the quick heuristic that we used to select it. We’ve provided more clarity around \beta and the specific GP kernels used. We show that any convex features can be used to construct a kernel which will admit a DC decomposition of the sample paths, and we simply take these features to be ReLUs (which have a corresponding arc-cosine limiting kernel).
>
> We used the same value of c for all methods, per objective function: for example, all experiments on 6D Rosenbrock used the same parameters, including EI and UCB. We have included some sensitivity experiments into the main paper.
>
> Theoretically, \beta must be increased to ensure sublinear cumulative regret (this is also a key point in Srinivas’ original UCB regret proof). We actually found that increasing \beta per iteration did not always give the best empirical results, depending on the objective function being optimised. And this phenomenon is common across all BO hyperparameters: we have theoretical support for various approaches, but in reality it is difficult to know in advance which hyperparameters are going to be most suited for which objective. We had details of which regimes we chose for \beta in the appendix, but have now moved discussion of this into the main paper. We want to be careful to not get too into the weeds regarding hyperparameter selection, as this is not the primary focus of the paper.
>
> While Wilson (2020) did not necessarily introduce a new method, they showed how pathwise updates via Matheron’s rule can be used to generate GP posterior sample paths. The separation between the prior sample and the posterior updates is critical for our method. You may consider our random Fourier features (RFF) experiments to be this, which is vanilla Thompson sampling using a squared exponential kernel (yellow lines on plots) and cosine features, with samples generated per Wilson (2020).
>
> We’ve specified the DC algorithm further in the manuscript. To answer your question here, the argmin subproblem can be solved using any local optimiser, and we used L-BFGS. The advantage is that this subproblem is convex. DCA can be likened to a vanilla Newton method: where a Newton method will approximate the objective with a quadratic around the incumbent point per iteration, DCA will approximate the objective using the convex component of the objective function itself per iteration.
>
> You’re right to see that the theory is not necessarily required. As is standard in BO regret proofs, we assume that the argmax of the acquisition can be found globally and exactly at each round - although this cannot be guaranteed in practice, which motivates this paper (amongst others) in finding better ways to solve the acquisition subproblem. The theory is novel in that it shows Thompson sampling regret bounds where samples paths are drawn according to Wilson (2020). What happens to regret bounds under inexact acquisition solutions is beyond the scope of this paper, although an important research direction.
>
> Yes, the only difference between the two is that ReLU (DC, ours) uses DCA to optimise the sample, whereas ReLU (no DC) will simply use L-BFGS, not exploiting the DC structure.
>
> We’ve added in some further discussion regarding your point on Figure 4. This was mainly used as some empirical justification to say that Matern kernels should not necessarily be the default choice of kernel, and that a GP surrogate under an arc-cosine kernel (with random ReLU features) may actually more closely model the data. Of course, BO performance is very sensitive to kernel and kernel hyperparameter choice. Our intention is to show some light evidence that despite DCTS requiring particular kernel types built from convex features, this does not necessarily make for a worse model.
>
> We’ve also addressed your minor change suggestions in the paper, including removing the redundant lemmas.

---

> > ### Comment · Reviewer_RxMQ · 2026-04-02
> >
> > Thank you for the discussion on c in the main paper and for providing an ablation study. Now, the hyperparameter c appears to have a very strong influence. How did you choose c in your experiments? The fact that you use 3 different values for 6 different functions suggests to me that the proposed method crucially relies on the correct setting of c.
> >
> > Following up on this, I am wondering if the bowl is only applied to the ReLU (DC, ours) or also the other methods?
> >
> > Figure 4 shows the squared exponential kernel, but Section 4.1 discusses the Matern kernel. How does this go together?

---

> > > ### Author Response · Authors · 2026-04-19
> > >
> > > Thank you for your comment. Yes, c does have a strong influence on the regret curve as is illustrated in Figure 5. This sensitivity curve would look similar for other hyperparameters such as kernel lengthscale, kernel variance and domain size. For our example cases, we experimented with various hyperparameter settings by hand.
> > >
> > > In order to fairly compare the methods, the bowl is applied each method in the regret plots, with c fixed per objective function.
> > >
> > > Thank you for your pickup with the reference to Matern kernels in section 4.1. Squared exponential kernel was indeed used for all experiments, and we have changed the language to make this more clear.

---

### Decision · Action_Editor_zKaB · 2026-06-29

**Recommendation:** Reject

**Additional Comments:**

I would like to apologize to the authors for the delay in making a recommendation. The delays in the process were unfortunate.

And I'd once again suggest they resubmit when ready.

**Audience:**

No

**Audience Explanation:**

Bayesian optimization is certainly of interest to a portion of the TMLR audience. However, from the reviews, I believe the yet-unknown limitations of the methodology (including the strong dependence on a hyperparameter) render the work, as-is, potentially not yet of interest.

However, I do believe this can be overcome in revision.

**Claims And Evidence:**

No

**Claims Explanation:**

This manuscript considers a new approach for drawing GP samples (or rather, a prior yielding samples with useful structure) for use in Thompson sampling / Bayesian optimization. The samples are constructed to belong to a class that makes them amenable to optimization, which is useful in the acquisition process.

During the initial review phase, the reviewers expressed interest in the methodology, but also identified a number of potential issues:

- strong dependence on a hyperparameter (and a lack of evidence investigating this dependence)
- some questions regarding the theoretical analysis
- some concerns regarding the design of the empirical study, including the choice of baselines

The author-reviewer discussion period was unfortunately (greatly) delayed due to some unfortunate circumstances. However, although the author responses did help to clear up some of these issues, the majority of the reviewers continue to have blocking concerns.

Quoting the official recommendations:

> * The overall method appears to overly rely on a prior that requires careful setting of a hyperparameter.
> * I acknowledge the theory and practice gap in Theorem 3.1 as the revision discusses it clearly. However, I think a stronger analysis incorporating approximate maximization of the Thompson sample would make the result more relevant to the DCA-based method.
> * In the comparison between DCA and L-BFGS, the iteration counts are not directly comparable, because each DCA outer iteration solves an inner optimization problem. The two approaches may therefore use different numbers of function/gradient evaluations. A more fair comparison on wall-clock time and numbers of function and gradient evaluations may be preferred.
> * Section 4 indicate that the performance is sensitive to the coefficient . It would be helpful to clarify how it is selected for each problem

I do believe these concerns (and those mentioned below) could be satisfactorily addressed by the authors with major revisions, and I encourage them to resubmit if desired.

**Resubmission Of Major Revision:**

The authors may consider submitting a major revision at a later time.